# Discrete viral E2 lysine residues and scavenger receptor MARCO are required for clearance of circulating alphaviruses

**Kathryn S Carpentier[1], Bennett J Davenport[1], Kelsey C Haist[1], Mary K McCarthy[1], Nicholas A May[1], Alexis Robison[2], Claudia Ruckert[2], Gregory D Ebel[2], Thomas E Morrison[1]\***

[1]Department of Immunology and Microbiology, University of Colorado School of Medicine, Aurora, United States; [2]Department of Microbiology, Immunology, and Pathology, Colorado State University, Fort Collins, United States

**Abstract** The magnitude and duration of vertebrate viremia is a critical determinant of arbovirus transmission, geographic spread, and disease severity. We find that multiple alphaviruses, including chikungunya (CHIKV), Ross River (RRV), and o'nyong 'nyong (ONNV) viruses, are cleared from the circulation of mice by liver Kupffer cells, impeding viral dissemination. Clearance from the circulation was independent of natural antibodies or complement factor C3, and instead relied on scavenger receptor SR-A6 (MARCO). Remarkably, lysine to arginine substitutions at distinct residues within the E2 glycoproteins of CHIKV and ONNV (E2 K200R) as well as RRV (E2 K251R) allowed for escape from clearance and enhanced viremia and dissemination. Mutational analysis revealed that viral clearance from the circulation is strictly dependent on the presence of lysine at these positions. These findings reveal a previously unrecognized innate immune pathway that controls alphavirus viremia and dissemination in vertebrate hosts, ultimately influencing disease severity and likely transmission efficiency.
DOI: https://doi.org/10.7554/eLife.49163.001

**\*For correspondence:**
thomas.morrison@ucdenver.edu

**Competing interests:** The authors declare that no competing interests exist.

## Introduction

During the past several decades, multiple arboviruses, including dengue (DENV), chikungunya (CHIKV), and Zika (ZIKV) viruses, have re-emerged to cause widespread epidemics affecting millions of people (*Weaver, 2018*). For example, DENVs are estimated to cause ~390 million infections per year resulting in ~500,000 cases of dengue hemorrhagic fever/dengue shock syndrome that can result in death (*Bhatt et al., 2013*). Since 2004, CHIKV has infected millions of people and expanded into Europe, Asia, the Pacific region, and the Americas (*Moro et al., 2010*; *Volk et al., 2010*; *Zeller et al., 2016*). Joint pain, swelling, and stiffness, and tenosynovitis, can endure for months to years after CHIKV infection (*Borgherini et al., 2008*; *Couturier et al., 2012*; *Gérardin et al., 2011*; *Rodriguez-Morales et al., 2016*; *Schilte et al., 2013*). As chronic CHIKV disease is debilitating, large epidemics have severe economic impact (*Cardona-Ospina et al., 2015*; *Soumahoro et al., 2011*; *Vijayakumar et al., 2013*). ZIKV caused outbreaks on Yap Island (*Duffy et al., 2009*) and islands of French Polynesia (*Musso et al., 2018*) in 2007 and 2013, respectively, before being introduced to the Americas (*Metsky et al., 2017*). Here, ZIKV caused nearly one million confirmed and suspected cases and became recognized as a cause of congenital neurodevelopmental defects (*Pierson and Diamond, 2018*). The lack of safe and effective vaccines and therapeutics limits prevention and treatment of many arboviral diseases.

The magnitude and duration of viremia in vertebrate hosts are important determinants of arboviral emergence, transmission efficiency, and geographic spread (*Weaver, 2018*). A subset of

**eLife digest** Viruses transmitted by blood-feeding insects, such as mosquitoes and ticks, cause serious human diseases. In recent years these viruses (also known as arboviruses) have re-emerged at an unprecedented scale, leading to global outbreaks of diseases such as Zika or chikungunya fever. The severity of these diseases and how easily they can be transmitted depends, in part, on the level of virus in the host's bloodstream following infection. The more viral particles present in the blood, the easier it is for other insects that bite the host to become infected and help spread the disease. Yet, the mechanisms that hosts use to control the amount of virus present in the blood and how long it persists are poorly understood.

To investigate this further, Carpentier et al. used a combination of molecular and genetic techniques to study how mice clear particles of arbovirus from their bloodstream. Surgically removing the spleen from infected mice revealed that this organ, which filters out unwanted or damaged materials from blood, is not required to clear some arbovirus particles. Carpentier et al. found that removing these arboviruses from the blood instead required Kupffer cells, a type of immune cell found in the liver.

Genetically manipulating mice so they no longer produced a protein on the surface of Kupffer cells known as MARCO revealed that this receptor is needed to clear chikungunya viral particles. When MARCO was genetically deleted this led to an increase in the number of viral particles in the mice's bloodstream, and allowed the virus to spread more rapidly throughout the bodies of the mice. Further experiments on three different types of arboviruses showed that in order to be cleared by MARCO, each of these viruses needed a lysine residue – one of the building blocks that makes up proteins – at defined positions within their protein sequence.

These findings reveal a previously unknown mechanism for how hosts remove arbovirus particles from their bloodstream. Future studies could use this information to identify new ways to control the transmission and reduce the severity of these viral diseases.

DOI: https://doi.org/10.7554/eLife.49163.002

arboviruses, including CHIKV, DENV, and ZIKV, are capable of sustained human-mosquito-human transmission cycles, allowing for rapid spread through urban environments. This is in part due to the capacity of these viruses to generate a magnitude and duration of viremia in humans sufficient to infect mosquito vectors and propagate the transmission cycle (*Weaver, 2018*). Moreover, for many arboviruses, viremia levels correlate with disease severity (*Chow et al., 2011*; *de St Maurice et al., 2018*; *Pozo-Aguilar et al., 2014*; *Vaughn et al., 2000*; *Waggoner et al., 2016*). Despite the critical role for viremia during arboviral infections, innate host defenses that determine the magnitude and duration of arboviral viremia in vertebrate hosts are poorly defined.

Phagocytic cells are strategically positioned within organs such as the spleen and liver to detect and capture circulating self and non-self molecules. In the murine spleen, blood released from terminal arterioles drains into the marginal zone, which separates the white and red pulp (*Lewis et al., 2019*). Within the marginal zone, tissue resident marginal zone (MZM) and metallophilic (MMM) macrophages surveil the blood for circulating antigens, apoptotic cells, debris, and pathogens (*Borges da Silva et al., 2015*; *Lewis et al., 2019*). In addition, splenic red pulp macrophages (RPM) also function to surveil the circulation by removing aberrant red blood cells as well as some pathogens, such as malaria parasites (*Borges da Silva et al., 2015*). Similar to the marginal zone of the spleen, cell populations in liver sinusoids, including tissue resident macrophages (i.e., Kupffer cells (KCs)), effectively survey blood as it percolates through the liver (*Hickey and Kubes, 2009*). For example, following intravenous inoculation, bacteria are captured within seconds by KCs (*Lee et al., 2010*; *Zeng et al., 2016*), whereas the absence of KCs results in persistent bacteremia (*Lee et al., 2010*). Although the surveillance function of these tissues and cells is well-established, how they influence viremia following arbovirus infections has not been defined.

In this study, we demonstrate that phagocytic cells are essential for efficient clearance of multiple alphaviruses from the circulation, thus limiting viremia and impeding viral dissemination. Experiments in splenectomized mice showed that the spleen is dispensable for clearance of these circulating alphaviruses. Consistent with this, we found that virus accumulates in the liver, and liver resident KCs

play a dominant role in removing circulating alphavirus particles. Mechanistically, we identified SR-A6 (MARCO) as the receptor required for efficient alphavirus clearance. We also found that discrete lysine (K) to arginine (R) mutations in the E2 glycoproteins of CHIKV and o'nyong 'nyong virus (ONNV) (E2 K200R), as well as Ross River virus (RRV) (E2 K251R), abrogated clearance of circulating alphavirus particles, and promoted rapid viral dissemination. Moreover, substitution of the lysine residues at CHIKV E2 200 or RRV E2 251 with a variety of other amino acids also allowed for clearance evasion, indicating an essential requirement for key lysine residues in E2 for efficient viral clearance from the circulation. These findings reveal a previously unrecognized innate immune pathway that controls alphavirus viremia and dissemination in vertebrate hosts.

## Results

### Phagocytic cells efficiently clear multiple alphaviruses from the circulation

To explore whether phagocytic cells participate in the clearance of circulating arboviruses, we treated mice intravenously (i.v.) with PBS- (control) or clodronate-loaded liposomes (PLL or CLL, respectively) to deplete phagocytic cells in the spleen, liver, and circulation that are in direct contact with blood (*Seiler et al., 1997*; *Van Rooijen and Sanders, 1994*). The uptake of CLL by phagocytic cells results in intracellular release of the encapsulated clodronate which is cytotoxic (*Van Rooijen, 1989*). At 42 hr (h) post-treatment, mice were i.v. inoculated with clinical isolates of CHIKV, ONNV, or RRV, all pathogenic alphaviruses (*Morrison, 2014*; *Suhrbier et al., 2012*), and viral RNA or infectious virus in the serum was analyzed at 45 minutes (min) post-inoculation. In PLL-treated mice, viral particles (CHIKV and RRV) or plaque forming units (ONNV) were nearly undetectable in the serum by 45 min post-inoculation (*Figure 1*). In stark contrast, CLL-treated mice had 3–4 orders of magnitude higher amounts of viral particles (CHIKV and RRV) or plaque-forming units (ONNV) in the serum at this time point (*Figure 1*). Viral RNA in PLL-treated mice also was absent from the clotted fraction of the blood (*Figure 1—figure supplement 1*), demonstrating that CHIKV particles had been removed

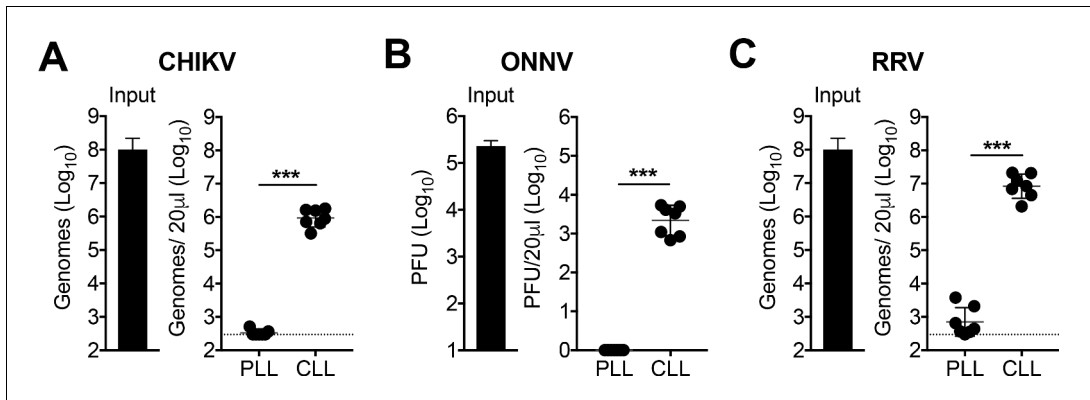

**Figure 1.** Phagocytic cells efficiently clear multiple alphaviruses from the circulation. (**A–C**) WT C57BL/6 mice were treated intravenously (i.v.) with PBS- (PLL) or clodronate-loaded liposomes (CLL). At 42 hr post-treatment, mice were inoculated i.v. with CHIKV (**A**), ONNV (**B**) or RRV (**C**), and viral genomes in the inoculum (input) and serum at 45 min post-inoculation were quantified by RT-qPCR (**A and C**) or plaque assay (**B**). Mean ± SD. N = 7, two experiments. Mann-Whitney test; ***p<0.001. *Figure 1—figure supplement 1* shows that viral RNA is undetectable in the clotted fraction of the blood.
DOI: https://doi.org/10.7554/eLife.49163.003

The following source data and figure supplement are available for figure 1:

**Source data 1.** Raw data for *Figure 1A-C*.
DOI: https://doi.org/10.7554/eLife.49163.005
**Figure supplement 1.** CHIKV RNA is undetectable in the clotted fraction of the blood at 45 min post-inoculation.
DOI: https://doi.org/10.7554/eLife.49163.004

from the circulation. These findings suggest that blood-exposed phagocytic cells, such as those in the spleen or liver, are required for clearance of blood-borne alphaviruses.

## Liver Kupffer cells clear CHIKV from the circulation

The spleen and liver contain phagocytic cell populations that are strategically positioned to capture circulating pathogens. To determine the relative contribution of the spleen, viral clearance was evaluated in mice that underwent sham or splenectomy surgeries. CHIKV was efficiently cleared from the circulation of splenectomized mice (*Figure 2A*), demonstrating that the spleen is not required for clearance of circulating CHIKV. Moreover, CLL-pretreatment blocked CHIKV clearance in splenectomized mice (*Figure 2B*), suggesting that the remaining phagocytic cells in the liver are sufficient to clear CHIKV from the circulation. Consistent with this, CHIKV RNA accumulated in the liver of control PLL-treated mice, and this specific accumulation was lost in CLL-treated mice (*Figure 2C*). In

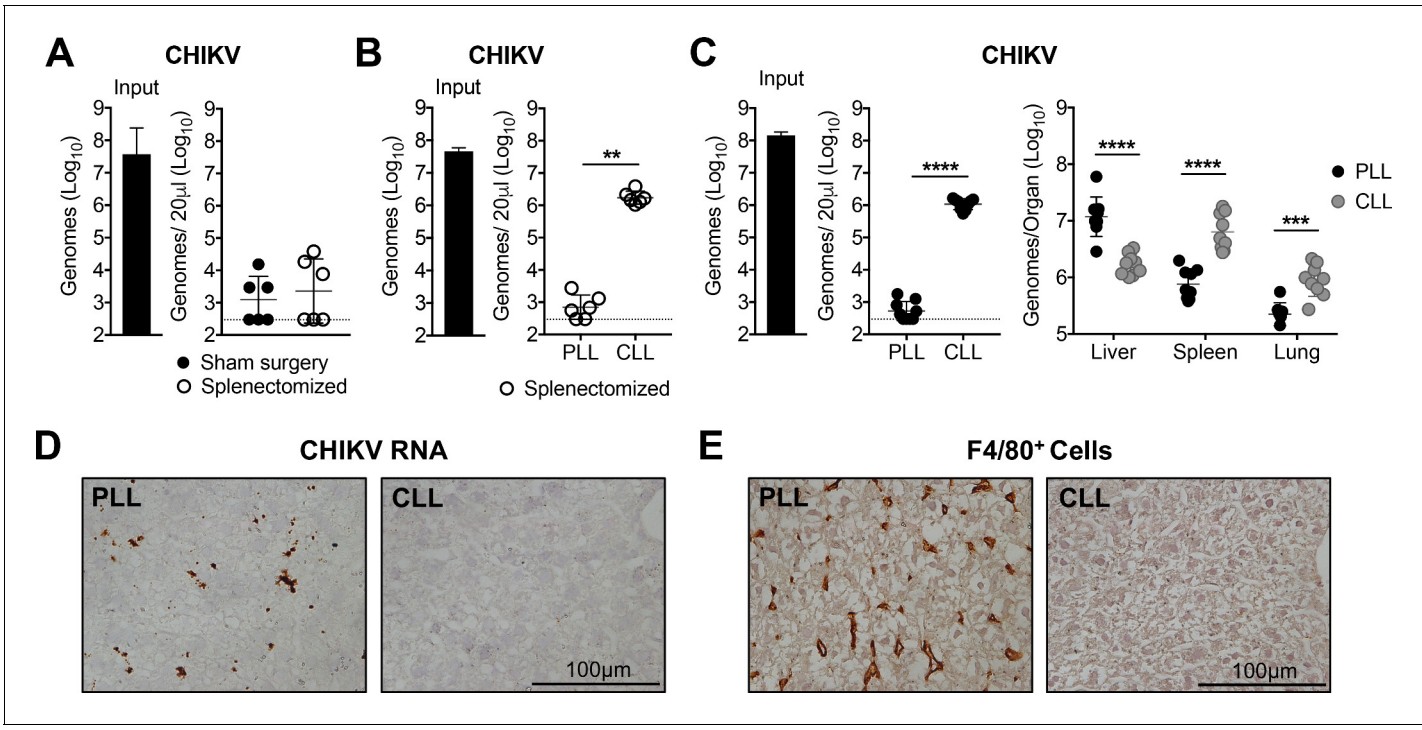

**Figure 2.** Liver Kupffer cells clear CHIKV from the circulation. (**A**) WT C57BL/6 mice that underwent a sham or splenectomy surgery were inoculated i.v. with CHIKV and viral genomes in the inoculum and serum at 45 min post-inoculation were quantified by RT-qPCR. Mean ± SD. N = 6, two experiments. Mann-Whitney test; p>0.05. (**B**) Splenectomized WT C57BL/6 mice were treated i.v. with PLL or CLL. At 42 hr post-treatment, mice were inoculated i.v. with CHIKV and viral genomes in the inoculum (input) and serum at 45 min post-inoculation were quantified by RT-qPCR. Mean ± SD. N = 6, two experiments. Mann-Whitney test; **p<0.01. (**C**) WT C57BL/6 mice were treated and inoculated as in (**B**), and viral genomes present at 45 min post-inoculation in the serum or indicated tissues were quantified by RT-qPCR. Mean ± SD. N = 9, two experiments. Mann-Whitney test or Two-tailed t-test; ***p<0.001, ****p<0.0001. (**D and E**) WT C57BL/6 mice were treated as in (**B**) and inoculated with CHIKV at 42 hr post-treatment. Livers were collected at 45 min post-inoculation, and RNA Scope in situ hybridization (**D**) or IHC (**E**) were performed to visualize viral RNA localization or F4/80+ macrophages, respectively. Brown staining is indicative of viral RNA (**D**) or F4/80+ macrophages (**E**). N = 6–7, two experiments. *Figure 2—figure supplement 1* shows representative images of CHIKV RNA Scope in situ hybridization and F4/80 IHC for all biological replicates, and *Figure 2—figure supplement 2* shows flow cytometry analysis of liver macrophages and dendritic cells (DCs) in PLL- and CLL-treated mice.
DOI: https://doi.org/10.7554/eLife.49163.006

The following source data and figure supplements are available for figure 2:

**Source data 1.** Raw data for *Figure 2A-C*.
DOI: https://doi.org/10.7554/eLife.49163.009
**Figure supplement 1.** CHIKV RNA and F4/80+ cells are detectable in the livers of PLL- but not CLL-treated mice.
DOI: https://doi.org/10.7554/eLife.49163.007
**Figure supplement 2.** Clodronate liposome treatment depletes liver macrophages, but has minimal impact on DCs.
DOI: https://doi.org/10.7554/eLife.49163.008

contrast, viral RNA levels increased in the spleen and lung of CLL-treated mice compared with control PLL-treated mice (*Figure 2C*), likely due to the high amounts of virus present in contaminating blood. Moreover, CHIKV RNA is readily detectable by in situ hybridization in the livers of PLL-treated mice at 45 min post-inoculation, but not in CLL-treated mice (*Figure 2D*). While both dendritic cells (DCs) and macrophages can sequester pathogens in the liver (*Jenne and Kubes, 2013*), treatment of mice with CLL resulted in depletion of F4/80$^+$ liver macrophages (*Figure 2E*, *Figure 2—figure supplement 1* and *Figure 2—figure supplement 2*) but had minimal impact on CD11c$^+$ DCs (*Figure 2—figure supplement 2*), suggesting macrophages are responsible for alphavirus clearance. Collectively, these data suggest that Kupffer cells (KCs), the tissue resident macrophage population in the liver (*Jenne and Kubes, 2013*), clear alphavirus particles from the circulation.

## The scavenger receptor MARCO is essential for the clearance of circulating alphavirus particles

Clearance of circulating bacteria and other particles from the circulation often involves opsonins such as antibodies and complement component 3 (C3) (*Guilliams et al., 2014*; *Kubes and Jenne, 2018*; *van Lookeren Campagne et al., 2007*). However, CHIKV and RRV particles were efficiently cleared from the circulation of C3$^{-/-}$ (*Figure 3A*) and B cell deficient μMT mice (*Figure 3B*), which lack C3 and circulating IgM and IgG, respectively (*Kitamura et al., 1991*; *Wessels et al., 1995*). These data suggest that clearance of circulating alphavirus particles is independent of receptors for fragments of C3 and of Fc receptors. Scavenger receptors (SRs) are a large family of transmembrane proteins that capture and eliminate a broad range of endogenous and microbial ligands via non-opsonic mechanisms (*Canton et al., 2013*; *Mukhopadhyay and Gordon, 2004*). To test the hypothesis that one or more SRs mediate clearance of alphavirus particles from the circulation, 5 min prior to inoculation of virus, mice were pre-treated i.v. with the polyanionic SR competitive inhibitor poly (I), or poly(C), a polyanion that is not a competitive inhibitor of SRs (*Pearson et al., 1993*). Similar to pretreatment with CLL, poly(I) pretreatment potently inhibited the clearance of circulating CHIKV and RRV particles (*Figure 3C*). Similar results were observed following pretreatment of mice with dextran sulfate (DS), another SR ligand (*Platt and Gordon, 1998*) (*Figure 3D*). To identify SRs that are expressed by KCs, we mined publicly available single-cell transcriptomic data of the mouse liver (*Tabula Muris Consortium et al., 2018*), and found SR-A1 (MSR1), SR-A6 (MARCO), SR-B1 (SCARB1), SR-B2 (CD36), and SR-I1 (CD163) mRNAs to be robustly expressed in KCs (*Figure 3—figure supplement 1*). Of these, only SR-A1 and MARCO are known to bind poly(I) and DS (*Chen et al., 2006*; *Platt and Gordon, 1998*). To investigate the relative contributions of SR-A1 and MARCO, CHIKV clearance was evaluated in SR-A1$^{-/-}$ and MARCO$^{-/-}$ mice. CHIKV particles were efficiently cleared from the circulation of SR-A1$^{-/-}$ mice, and this clearance was inhibited by pre-treatment of SR-A1$^{-/-}$ mice with poly(I) (*Figure 3E*), indicating that CHIKV clearance was still occurring through a SR-dependent mechanism. In contrast, CHIKV particles remained readily detectable in the serum of MARCO$^{-/-}$ mice at 45 min post-inoculation (*Figure 3F*). Moreover, whereas pretreatment with CLL blocked clearance of circulating CHIKV particles in WT mice, this treatment had no effect on the level of CHIKV in the circulation of MARCO$^{-/-}$ mice (*Figure 3G*). Furthermore, CHIKV RNA was detectable by in situ hybridization in the liver of PLL-treated WT mice, but not in the liver of PLL-treated MARCO$^{-/-}$ mice or CLL-treated WT and MARCO$^{-/-}$ mice (*Figure 3H*). Despite the virological differences in the circulation and liver of PLL-treated WT and MARCO$^{-/-}$ mice, these mice had a similar number of KCs in the liver (*Figure 3I*). Finally, both WT and MARCO$^{-/-}$ KCs were efficiently depleted by CLL (*Figure 3H and I*), suggesting that MARCO$^{-/-}$ KCs retain phagocytic capacity. Collectively, these findings demonstrate that MARCO is essential for clearance of CHIKV particles from the circulation.

## CHIKV E2 K200R evades phagocytic cell-mediated clearance

We next investigated which viral features promote clearance of alphaviral particles from the circulation. In prior studies, DENV and West Nile virus (WNV) clearance from the circulation of mice was found to be influenced by N-linked glycans on viral envelope glycoproteins (*Fuchs et al., 2010*). CHIKV E2 is predicted to be glycosylated at N345, N263 and N273 (*Metz et al., 2011*; *Voss et al., 2010*). N345 is located in the E2 stem near the transmembrane domain, making it unlikely that glycans at this position mediate interactions with MARCO (*Voss et al., 2010*). To investigate if the

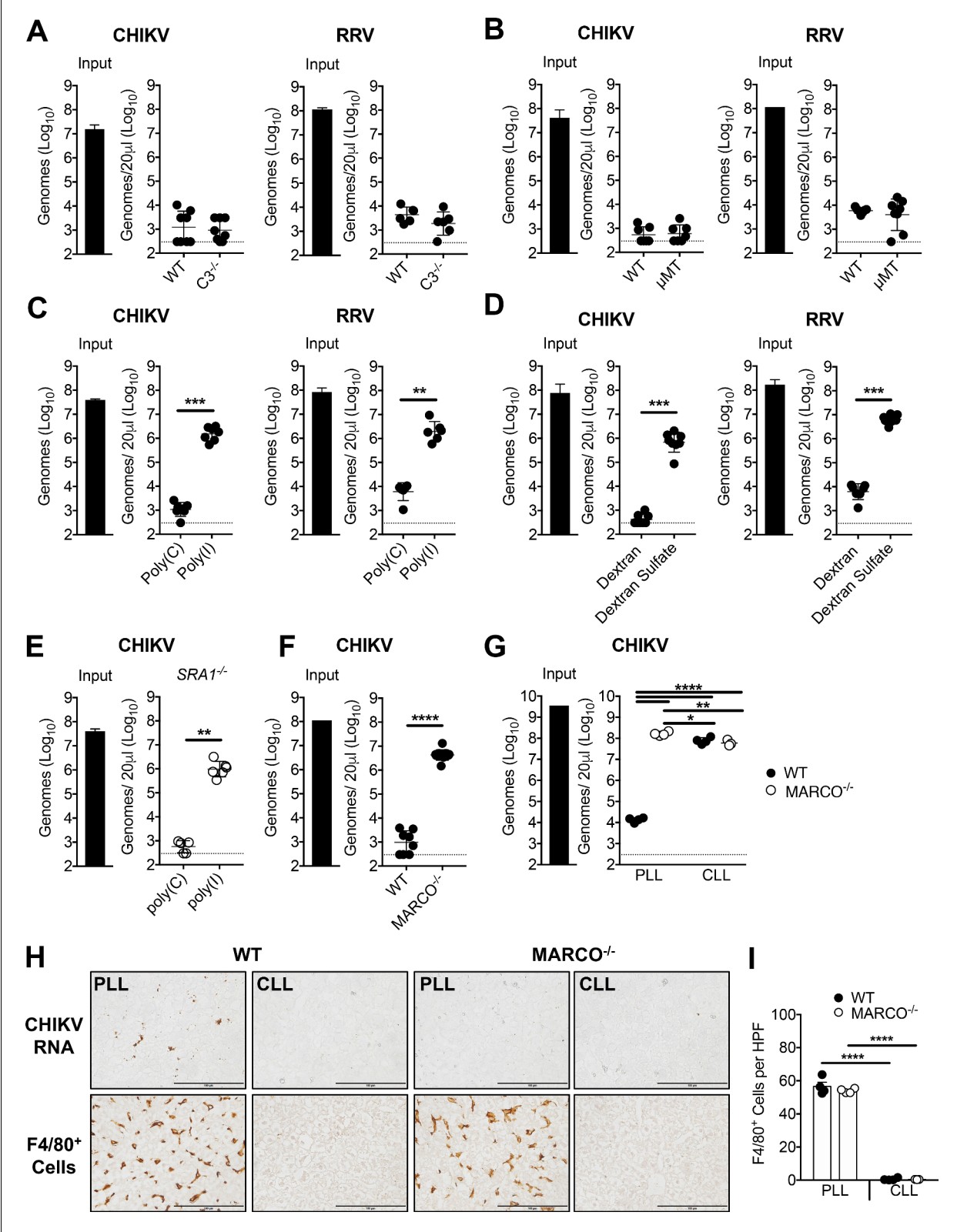

**Figure 3.** The scavenger receptor MARCO is required for clearance of CHIKV and RRV. (**A**) WT or C3[-/-] C57BL/6 mice were inoculated i.v. with the indicated virus, and viral genomes in the inoculum (input) and serum at 45 min post-inoculation were determined by RT-qPCR. Mean ± SD. N = 6–8, two experiments. Mann-Whitney test; p>0.05. (**B**) WT or μMT C57BL/6 mice were inoculated i.v. with the indicated virus, and analyzed as in (**A**). Mean ± SD. N = 7–8, two experiments. Mann-Whitney test; p>0.05. (**C**) WT C57BL/6 mice were treated i.v. with 200 μg of poly(I) or poly(C) 5 min prior

*Figure 3 continued on next page*

*Figure 3 continued*

to inoculation with the indicated virus. Viral genomes in the serum at 45 min post-inoculation were quantified by RT-qPCR. Mean ± SD. N = 6–7, two experiments. Mann-Whitney test or Two-tailed unpaired t-test; **p<0.01, ***p<0.001. (D) WT C57BL/6 mice were treated i.v. with 200 µg of dextran or dextran sulfate 5 min prior to inoculation with the indicated virus, and analyzed as in (C). Mean ± SD. N = 7–8, two experiments. Mann-Whitney test or Two-tailed unpaired t-test; ***p<0.001. (E) SR-A1$^{-/-}$ mice were treated, incoculated, and evaluated as in (C). Mean ± SD. N = 5–6, two experiments. Mann-Whitney test; **p<0.01. (F) WT or MARCO$^{-/-}$ C57BL/6 mice were inoculated and evaluated as in (A). Mean ± SD. N = 8–11, two experiments. Mann-Whitney test; ****p<0.0001. (G–I) WT and MARCO$^{-/-}$ C57BL/6 mice were treated i.v. with PBS- (PLL) or clodronate-loaded liposomes (CLL). At 42 hr post-treatment, mice were inoculated i.v. with CHIKV. (G) Viral genomes in the inoculum (input) and serum at 45 min post-inoculation were quantified by RT-qPCR. Mean ± SD. N = 3–4. One-way ANOVA with Tukey's multiple comparisons test; *p<0.05, **p<0.01, ****p<0.0001. (H) Livers were collected at 45 min post-inoculation, and RNA Scope in situ hybridization or IHC were performed to visualize viral RNA localization or F4/80$^+$ macrophages, respectively. Brown staining is indicative of viral RNA or F4/80$^+$ macrophages. (I) F4/80 positive cells in 10 randomly selected high power fields (HPF) per section were counted in a blinded manner and used to calculate the average number of F4/80 positive cells per field. Mean ± SD. N = 3–4. One-way ANOVA with Tukey's multiple comparisons test; ****p<0.0001. *Figure 3—figure supplement 1* shows the mRNA expression of various scavenger receptors in murine liver cell subsets.

DOI: https://doi.org/10.7554/eLife.49163.010

The following source data and figure supplement are available for figure 3:

**Source data 1.** Raw data for *Figure 3A-G, I*.

DOI: https://doi.org/10.7554/eLife.49163.012

**Figure supplement 1.** Expression of various scavenger receptors by murine liver cell subsets.

DOI: https://doi.org/10.7554/eLife.49163.011

more surface exposed glycosylation sites influenced clearance, we introduced asparagine to glutamine mutations at N263 and N273 to disrupt glycosylation. Mutation of positions N263 or N273 alone, or in combination, had no effect on the clearance of CHIKV particles from the circulation (*Figure 4—figure supplement 1*), demonstrating that clearance of CHIKV particles does not rely on glycosylation at these sites.

Previously, we identified a mutation in the CHIKV E2 glycoprotein (E2 K200R) in virus persistently circulating in *Rag1*$^{-/-}$ mice (*Hawman et al., 2017*). This mutation enhanced CHIKV dissemination in mice following subcutaneous inoculation and caused more severe disease outcomes. These effects were associated with a strongly elevated viremia. Having identified an important role for phagocytic cells and MARCO in the clearance of CHIKV particles from the circulation, we hypothesized that the E2 K200R mutation allows for viral escape from phagocytic cell-mediated clearance. Indeed, unlike WT CHIKV, which is efficiently cleared from the circulation, CHIKV E2 K200R particles remained stably detectable in the serum for at least 1 hr post-inoculation (*Figure 4A*). Importantly, the differential clearance was not dependent on the cell type from which the viral particles were derived, as differential clearance of WT and E2 K200R CHIKV particles was maintained when virions were derived from human fibroblasts, a clinically relevant cell type (*Couderc et al., 2008*) (*Figure 4B*). The E2 K200R mutation did not impact the stability of viral particles in serum, demonstrating that this effect was not due to differential stability of WT and E2 K200R CHIKV particles (*Figure 4—figure supplement 2A*). Moreover, the levels of E2 K200R CHIKV particles in control PLL-treated mice were indistinguishable from WT and E2 K200R CHIKV particles in the circulation of mice pre-treated with CLL (*Figure 4C*), demonstrating that CHIKV E2 K200R allows for evasion of phagocytic cell-mediated clearance. The ability of E2 K200R to evade this response was not limited to WT C57BL/6J mice (*Figure 4C*), as evasion of clearance also was observed in 129S1/SvlmJ, and BALB/cJ mouse strains (*Figure 4—figure supplement 2B*).

To test if the effects of the E2 K200R mutation were unique to the CHIKV strain tested (AF15561; Asian genotype) or functioned within broader viral genetic contexts, we introduced an E2 K200R mutation into the genome of CHIKV strains representative of all phylogenetic lineages: Asian-American (strain 99659; *Jones et al., 2017*; *Lanciotti and Valadere, 2014*), East, Central, South African (strain SL15649; *Morrison et al., 2011*), and West African (strain 37997; *Tsetsarkin et al., 2006*; *Vanlandingham et al., 2005a*). In addition, we introduced an E2 K200R mutation into the genome of the SG650 strain of ONNV (*Lanciotti et al., 1998*). Similar to CHIKV AF15561, all WT strains of CHIKV and ONNV were cleared from the circulation at 45 min post-inoculation, whereas the levels of all E2 K200R mutant particles were substantially higher (*Figure 4D*). These results demonstrate

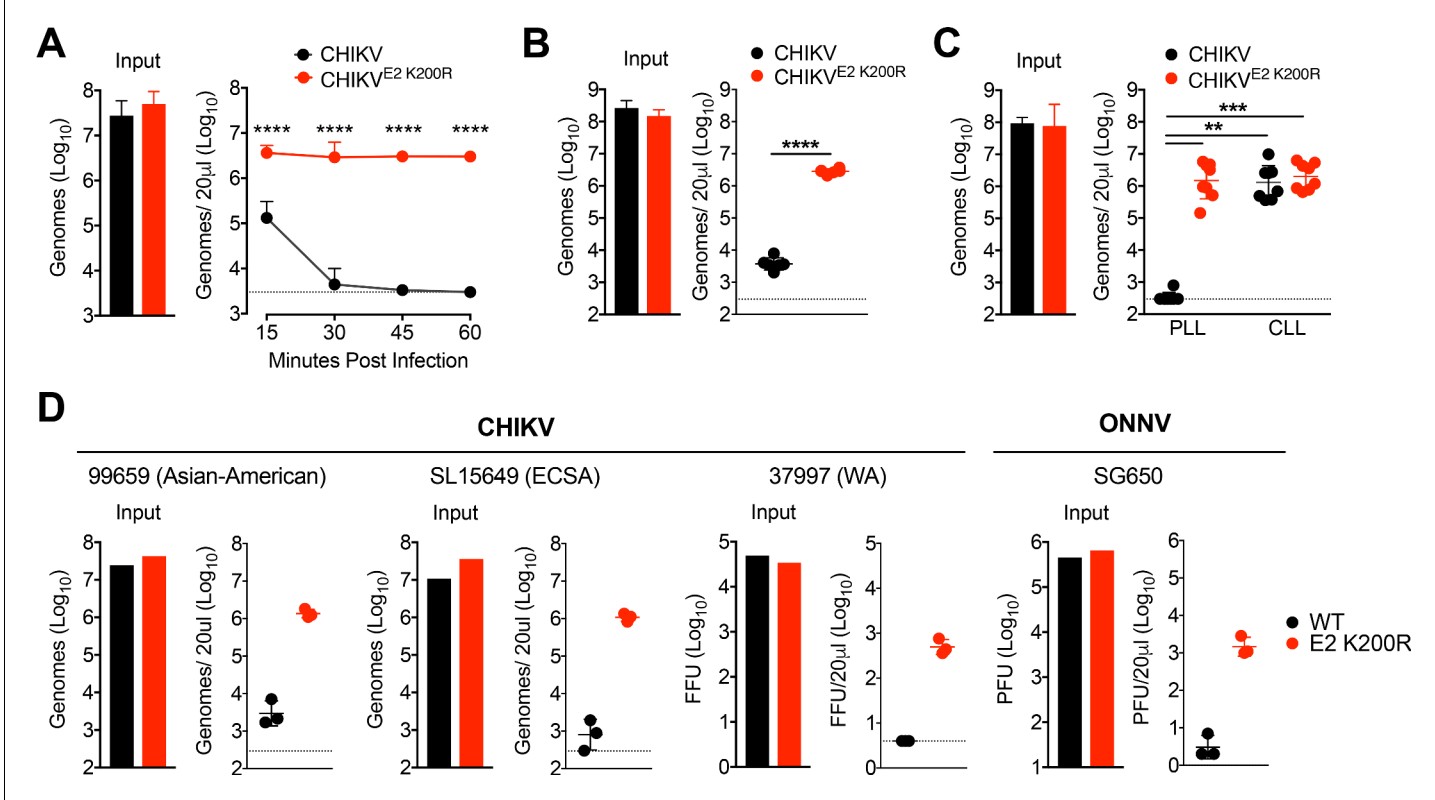

**Figure 4.** CHIKV E2 K200R evades phagocytic cell-mediated clearance. (**A**) WT C57BL/6 mice were inoculated i.v. with CHIKV or CHIKV E2 K200R, and viral genomes in the inoculum and serum at the indicated times post-inoculation were evaluated by RT-qPCR. Mean ± SD. N = 6 per time point, two experiments. Two-way ANOVA with Bonferroni's correction; ****p<0.0001. (**B**) WT C57BL/6 mice were inoculated with human fibroblast-derived CHIKV or CHIKV E2 K200R, and viral genomes in the inoculum and in the serum at 45 min post-inoculation were quantified by RT-qPCR. Mean ± SD. N = 6, two experiments. Two-tailed, unpaired t-test; ****p<0.0001. (**C**) WT C57BL/6 mice were treated i.v. with PBS liposomes or clodronate liposomes 42 hr prior to i.v. inoculation with CHIKV or CHIKV E2 K200R. Viral genomes in the inoculum and serum at 45 min post-inoculation were determined by RT-qPCR. Mean ± SD. N = 8, two experiments. Kruskal-Wallis; **p<0.01, ***p<0.001 (**D**) WT C57BL/6 mice were inoculated i.v. with the indicated viruses, and viral genomes in the inoculum and in the serum at 45 min post-inoculation were quantified by RT-qPCR (99659, SL15649), focus forming assay (37997), or plaque assay (ONNV SG650). Mean ± SD. N = 3, one experiment. Mann-Whitney test; p>0.05. *Figure 4—figure supplement 1* shows that mutation of E2 glycosylation sites does not influence CHIKV clearance from the circulation. *Figure 4—figure supplement 2* shows that the E2 K200R mutation does not alter the thermostability of CHIKV particles in serum (**A**), and that the E2 K200R mutation allows CHIKV particles to evade clearance from the circulation in multiple distinct mouse strains (**B**).

DOI: https://doi.org/10.7554/eLife.49163.013

The following source data and figure supplements are available for figure 4:

**Source data 1.** Raw data for *Figure 4A-D*.
DOI: https://doi.org/10.7554/eLife.49163.016
**Figure supplement 1.** Mutation of putative CHIKV glycosylation sites has no impact on viral clearance from the circulation.
DOI: https://doi.org/10.7554/eLife.49163.014
**Figure supplement 2.** CHIKV E2 K200R does not impact virus stability and evades clearance in multiple mouse strains.
DOI: https://doi.org/10.7554/eLife.49163.015

that an E2 K200R mutation broadly facilitates evasion of phagocytic-cell mediated clearance in multiple CHIKV strains, as well as the related ONNV.

## An E2 K200R mutation or deletion of MARCO allows for enhanced viremia and more rapid viral dissemination

To evaluate if phagocytic-cell mediated clearance of blood-borne virus has implications following a more natural route of infection, we inoculated mice subcutaneously with WT or E2 K200R viruses and evaluated viral burdens in various tissues at 1 d post-inoculation. Viral burdens in the serum were significantly higher in mice inoculated with E2 K200R strains (*Figure 5A*), indicating that

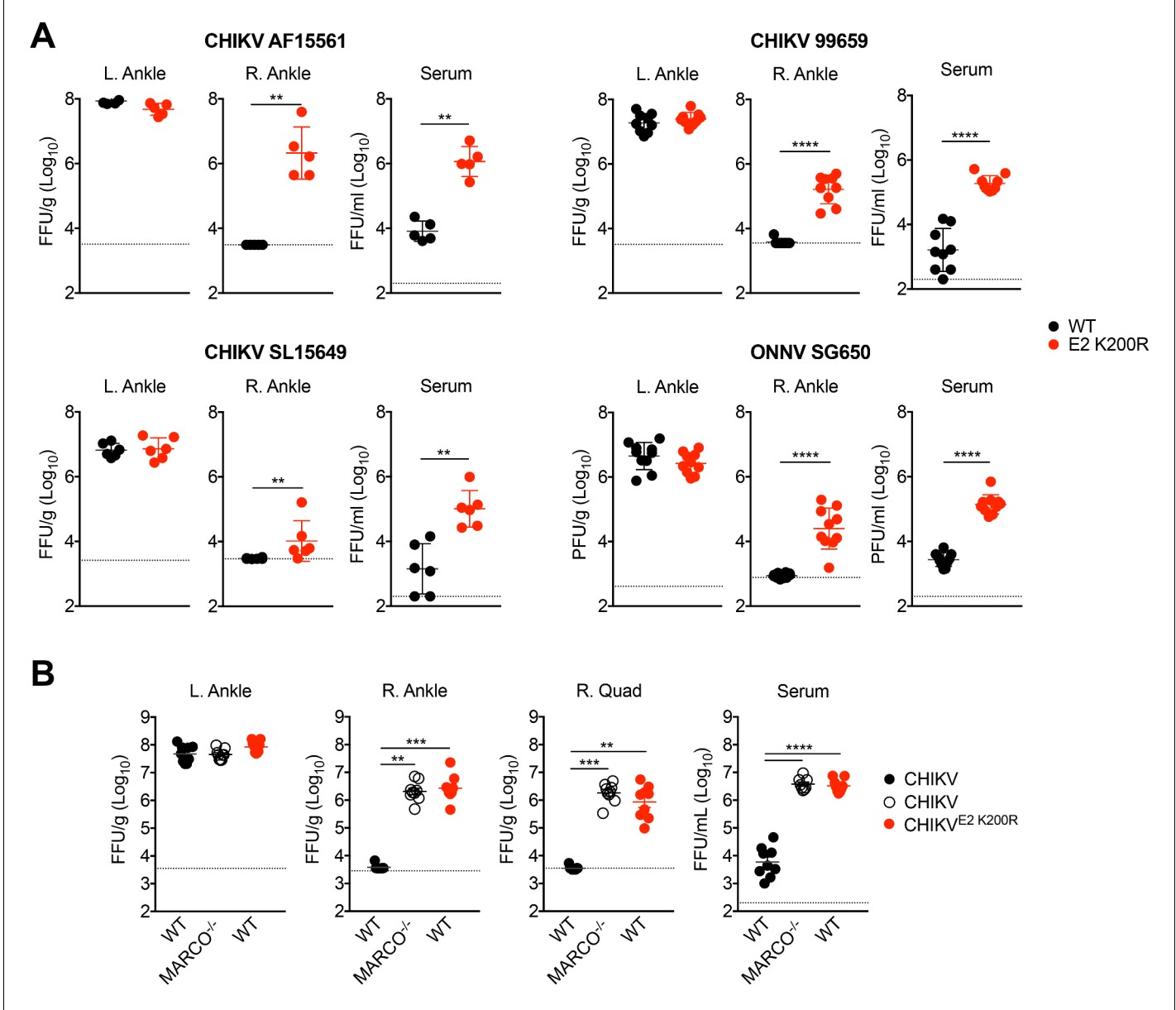

**Figure 5.** An E2 K200R mutation or deletion of MARCO allows for enhanced viremia and more rapid viral dissemination. (**A**) WT C57BL/6 mice were inoculated in the left footpad with 1,000 PFU of the indicated virus. At 24 hpi, infectious virus in the ipsilateral left ankle, contralateral right ankle, and serum were quantified by focus formation assay (CHIKV) or plaque assay (ONNV). Mean ± SD. N = 5–10, two experiments. Mann-Whitney test; **p<0.01, ****p<0.0001. (**B**) WT or MARCO[-/-] C57BL/6 mice were inoculated as in (**A**) with CHIKV or CHIKV E2 K200R. At 24 hpi, infectious virus in the ipsilateral left ankle, contralateral right ankle, right quadriceps, and serum were quantified by focus formation assay. Mean ± SD. N = 9, two experiments. Kruskal-Wallis or One-way ANOVA; **p<0.01, ***p<0.001, ****p<0.0001.

DOI: https://doi.org/10.7554/eLife.49163.017

The following source data is available for figure 5:

**Source data 1.** Raw data for *Figure 5A-B*.

DOI: https://doi.org/10.7554/eLife.49163.018

phagocytic cell-mediated clearance is an important immune mechanism to control viremia following subcutaneous inoculation. Moreover, although viral burdens were similar in tissues proximal to the site of inoculation (i.e, the L. ankle), significantly elevated levels of virus were detected in the contralateral right ankle of mice infected with E2 K200R strains (*Figure 5A*), demonstrating that this mutation facilitates more rapid viral dissemination.

To investigate if expression of MARCO also influences viremia and viral dissemination following subcutaneous inoculation of virus, we inoculated MARCO$^{-/-}$ mice subcutaneously with WT CHIKV and evaluated viral burdens at 1 d post-inoculation. Similar to infection of WT mice with CHIKV E2 K200R, infection of MARCO$^{-/-}$ mice with WT CHIKV resulted in an elevated viremia and more rapid viral dissemination to distal sites including the right ankle and quadriceps compared with infection of WT mice (*Figure 5B*). Collectively, these findings indicate that MARCO-dependent clearance of circulating CHIKV particles controls viremia and impedes viral dissemination.

## CHIKV E2 K200R does not impact vector competence or viral fitness in mosquitoes

As arboviruses, alphaviruses must maintain efficient replication in both vertebrate hosts and mosquito vectors. Given this, we evaluated whether the E2 K200R mutation influenced viral replication and/or dissemination in *Ae. aegypti* mosquitoes. Following blood-feeding inoculation of mosquitoes, similar levels of WT CHIKV and E2 K200R were detected in the bodies, legs and saliva of infected mosquitoes at 3 d post-infection, suggesting that in contrast to mice, the E2 K200R mutation had no impact on viral dissemination in mosquitoes (*Figure 6A*). To more rigorously test if this mutation affects viral fitness in mosquitoes, we performed in vitro and in vivo competition experiments with a WT CHIKV strain genetically marked with an ApaI restriction site (CHIKV-ApaI). Control 1:1 competitions of CHIKV and CHIKV-ApaI in C6/36 cells (*Figure 6B*) and microinjected *Ae. Aegypti* mosquitoes (*Figure 6C and D*) demonstrated that the genetic marker does not influence viral fitness. Similar results were observed following direct 1:1 competition of CHIKV and CHIKV E2 K200R, (*Figure 6B, C and D*), suggesting that the E2 K200R mutation has no selective disadvantage to WT virus in the mosquito vector. Collectively, these findings suggest that while the E2 K200R mutation dramatically influences CHIKV dissemination in the vertebrate host, it is neutral in the mosquito vector.

## RRV clearance from the circulation is mediated by a distinct lysine residue, E2 K251

The critical lysine at position E2 200 of CHIKV and ONNV is not conserved within RRV strains, which led us to investigate viral features that promote clearance of circulating RRV particles. In addition to RRV strain SN11 (*Figure 1C*), a 2009 clinical isolate (*Liu et al., 2011*), we found that RRV strain DC5692 (*Jupille et al., 2011*; *Lindsay, 1996*) also was efficiently cleared from the circulation by phagocytic cells (*Figure 7A*). In contrast, RRV strain T48 (*Doherty and Whitehead, 1963*) was resistant to clearance, and clearance was relatively unaffected by prior depletion of phagocytic cells (*Figure 7A*). Through chimeric virus analysis of DC5692 and T48 strains, we mapped the genetic determinant of RRV susceptibility to clearance to a region of the genome encoding a portion of capsid, E3 and E2 (*Figure 7—figure supplement 1A*). Within the E2 glycoprotein, the T48 and DC5692 RRV strains differ by only six amino acids (*Figure 7B*). Remarkably, mutating position E2 251 of T48 from arginine to lysine (R251K) was sufficient to make this strain more susceptible to clearance, while the corresponding K251R mutation in DC5692 conferred resistance to clearance (*Figure 7B*). Moreover, sequencing analysis confirmed the presence of a lysine at E2 251 in RRV strain SN11. The R251K mutation had no impact on the stability of RRV T48 in serum (*Figure 7—figure supplement 1B*). In addition, inoculation of purified viral particles generated in serum free media maintained the differential clearance of T48 and T48 E2 R251K, indicating that viral clearance is not mediated by factors in fetal bovine serum (*Figure 7—figure supplement 1C*). Collectively, these findings demonstrate that, in three independent alphaviruses, lysine to arginine mutations at distinct sites in the E2 glycoprotein (*Figure 7C*) allow for viral escape from phagocytic cell-mediated clearance.

## Lysine residues at E2 200 or E2 251 are essential for CHIKV and RRV clearance, respectively

The impact of lysine to arginine mutations was surprising as this is a conservative amino acid substitution. To further map biochemical features of amino acid side chains that associate with efficient viral clearance by phagocytic cells, we introduced a variety of amino acid substitutions at CHIKV E2 200. The amino acid panel was designed to evaluate positively charged (K, R, H), negatively charged (D), hydrophobic (A, L), and polar, uncharged (Q, S) amino acids (*Figure 8A*). In addition, the amino

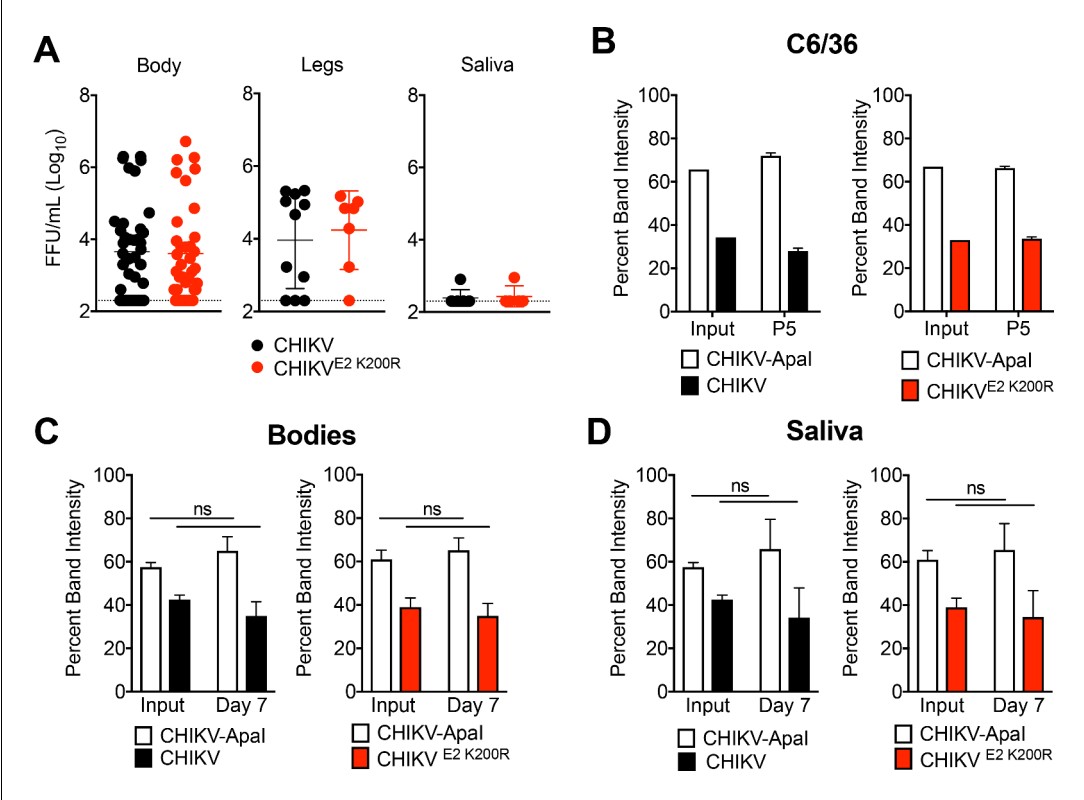

**Figure 6.** CHIKV E2 K200R has no impact on vector competence or viral fitness in mosquitoes. (**A**) *Ae. aegypti* mosquitoes were fed a blood meal containing $1.1 \times 10^6$ PFU/mL of CHIKV or CHIKV E2 K200R, and the head, legs, and saliva were collected at three dpi. Samples initially found to be positive for virus were evaluated by focus formation assay to quantify infectious virus. Mean ± SD. N = 50, one experiment. Mann-Whitney test; p>0.05. (**B**) *Ae. albopictus* C6/36 cells were infected in triplicate at an MOI of 1 PFU/cell with a 1:1 mixture of CHIKV marked with an ApaI restriction site (CHIKV-ApaI) and WT CHIKV or CHIKV E2 K200R, and 1/10th of the supernatant was serially passaged onto new C6/36 cells every 24 hr. RNA was extracted from the inoculum and supernatant of passage 5, cDNA was generated, and PCR amplified. Digestion of the PCR product was used to identify ratios of ApaI marked to unmarked virus, and the percent band intensity is displayed. Mean ± SD. N = 3, one experiment. (**C and D**) *Ae. aegypti* mosquitoes were microinjected with 138 PFU of 1:1 mixtures of CHIKV-ApaI and CHIKV, or CHIKV-ApaI and CHIKV E2 K200R. Bodies (**C**) and saliva (**D**) were collected at seven dpi. Ratios of each virus present in the input and in samples at day seven were evaluated as described in (**B**). Mean ± SD. N = 20, one experiment. Two-way ANOVA with Bonferroni's correction; p>0.05 for all comparisons.

DOI: https://doi.org/10.7554/eLife.49163.019

The following source data is available for figure 6:

**Source data 1.** Raw data for *Figure 6A-D*.

DOI: https://doi.org/10.7554/eLife.49163.020

acids selected varied in side chain length (*Figure 8A*). Remarkably, all substitutions tested prevented efficient clearance of circulating CHIKV particles following i.v. inoculation (*Figure 8B*). Moreover, a subset of these mutations was introduced at position E2 251 of RRV, and again all substitutions away from lysine allowed for RRV escape from phagocytic cell-mediated clearance (*Figure 8C*). These findings suggest that clearance of these alphavirus particles from the circulation is strictly dependent on the presence of a lysine residue at E2 200 (CHIKV) or E2 251 (RRV). The critical lysine residue that mediates clearance in RRV (E2 K251) is conserved in CHIKV (E2 K252). To investigate if this lysine residue also contributes to CHIKV clearance, we introduced an E2 K252R mutation. However, we found that CHIKV E2 K252R remained susceptible to clearance by phagocytic cells (*Figure 8D*), suggesting that, in contrast to RRV, a lysine at this position in E2 is not required for CHIKV clearance. Collectively, these findings demonstrate that specific lysine residues in the E2 glycoproteins of CHIKV, ONNV, and RRV facilitate phagocytic cell-mediated clearance of viral particles from the circulation of a vertebrate host.

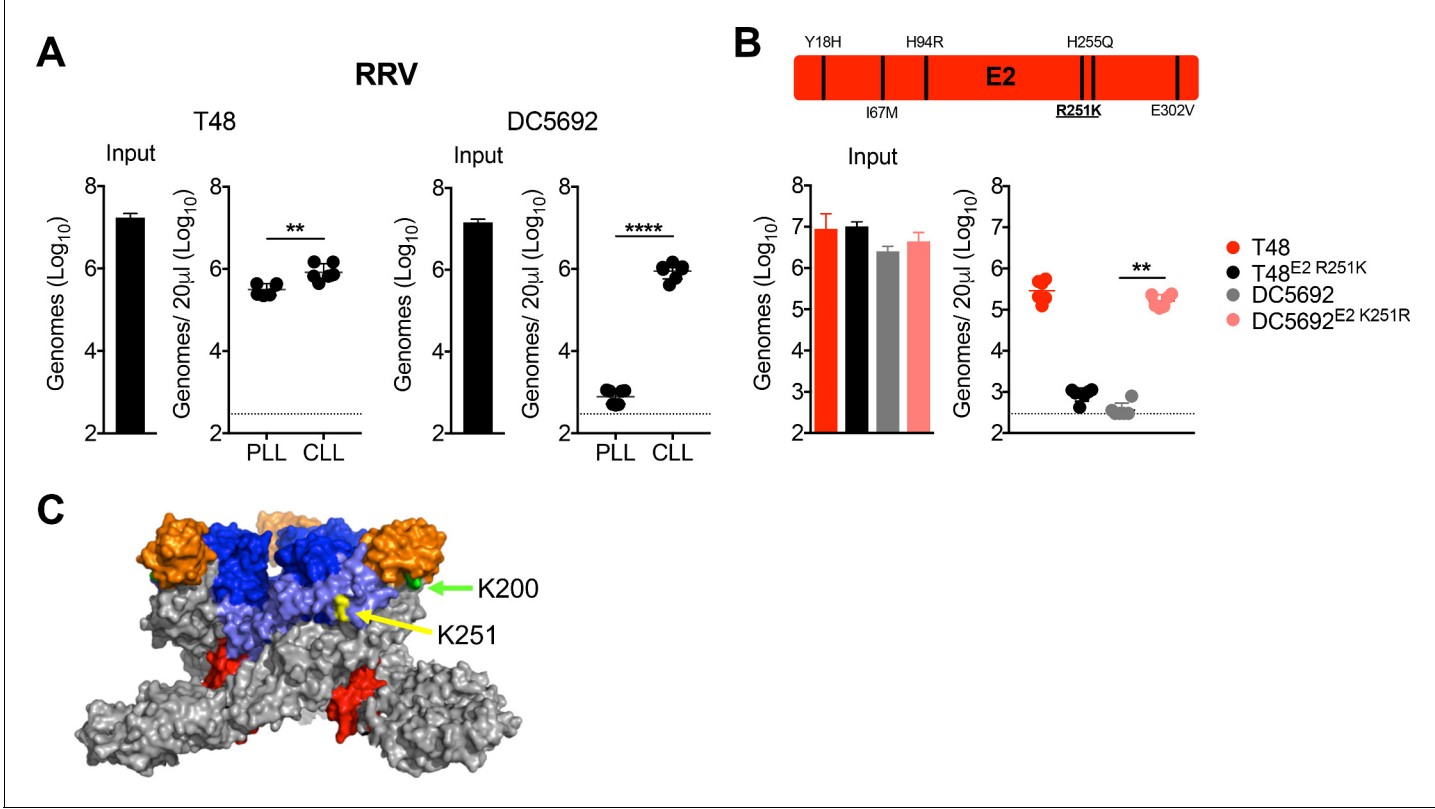

**Figure 7.** RRV clearance from the circulation is mediated by a distinct lysine residue, E2 K251. (**A**) WT C57BL/6 mice were treated i.v. with PLL or CLL 42 hr prior to i.v. inoculation with RRV strains T48 or DC5692. Viral genomes in the inoculum and in the serum at 45 min post-inoculation were quantified by RT-qPCR. Mean ± SD. N = 6, two experiments. Two-tailed unpaired t-test; **p<0.01, ****p<0.0001. (**B**) Schematic representation of the six amino acid differences between T48 and DC5649 within E2 shown, with position 251 underlined. WT C57BL/6 mice were inoculated i.v. with the indicated viruses and mutants. Viral genomes in inoculum and in the serum at 45 min post-inoculation were quantified as in (**A**). Mean ± SD. N = 6, two experiments. Kruskal-Wallis; **p<0.01. (**C**) CHIKV trimeric spike (PDB 3J2W), with E2 K200 highlighted in green, and E2 K251 (RRV numbering) highlighted in yellow. The E1 glycoprotein is shown in gray, E2 domain A is shown in blue, E2 domain B is shown in orange, E2 domain C is shown in red, and the β-ribbon connector is shown in light blue. *Figure 7—figure supplement 1* shows evaluation of the clearance of RRV T48 and RRV DC5692 structural gene chimeras (**A**), demonstrates that the E2 K251R mutation does not alter the thermostability of RRV particles in serum (**B**), and that the clearance phenotypes of RRV T48 and RRV T48 E2 R251K viral particles are maintained when purified viral particles are used for mouse inoculation studies (**C**).

DOI: https://doi.org/10.7554/eLife.49163.021

The following source data and figure supplement are available for figure 7:

**Source data 1.** Raw data for *Figure 7A-C*.
DOI: https://doi.org/10.7554/eLife.49163.023
**Figure supplement 1.** Chimeric analysis of T48 and DC5692, stability of T48 versus T48 E2 R251K, and serum clearance of purified viral particles.
DOI: https://doi.org/10.7554/eLife.49163.022

## Discussion

Our findings reveal a previously uncharacterized innate immune pathway that rapidly clears a number of distinct alphavirus particles from the circulation. While liver KCs have been implicated as a critical innate defense against blood-borne bacterial pathogens (*Jenne and Kubes, 2013*; *Krenkel and Tacke, 2017*; *Kubes and Jenne, 2018*), this work expands their role to additionally serve as sentinels for circulating CHIKV, ONNV, and RRV. Mechanistically, this innate clearance pathway occurs independently of opsonization with natural antibody or fragments of C3, and instead is mediated by the scavenger receptor MARCO. While our data suggest that KCs in the liver are primarily responsible for clearance of these alphavirus particles from the circulation, MARCO is known to be expressed by other macrophage populations exposed to the blood, including MZM in the spleen (*Borges da Silva et al., 2015*). Although we found that the spleen is not necessary for the

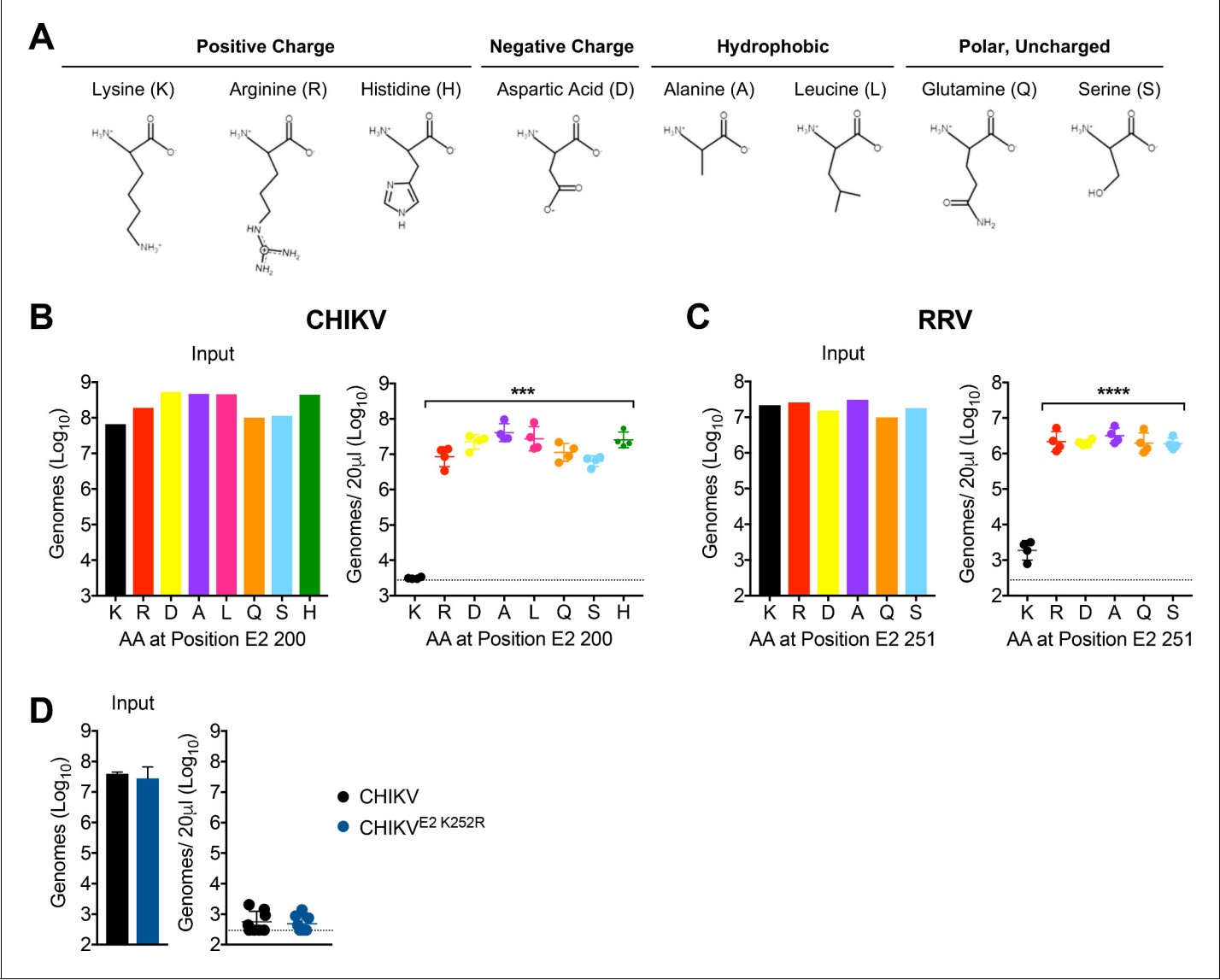

**Figure 8.** Lysine residues at E2 200 or E2 251 are essential for CHIKV and RRV clearance, respectively. (**A**) Properties of the amino acid side chains selected for substitution analysis at E2 K200 or K251. Amino acid structures generated using PepDraw. (**B**) WT C57BL/6 mice were inoculated i.v. with a panel of CHIKV mutants with different amino acid substitutions at position E2 200. Viral genomes in the inoculum and in the serum at 45 min post-inoculation were quantified by RT-qPCR. Mean ± SD. N = 4, one experiment. Kruskal-Wallis, comparing each group to WT virus containing K; ***p<0.001. (**C**) WT C57BL/6 mice were inoculated i.v. with a panel of RRV mutants with different amino acid substitutions at position E2 251. Viral genomes in the inoculum and in the serum at 45 min post-inoculation were quantified by RT-qPCR. Mean ± SD. N = 4, one experiment. One-way ANOVA, comparing each group to virus containing K; ****p<0.0001. (**D**) WT C57BL/6 mice were inoculated i.v. with CHIKV or CHIKV E2 K252R, and viral genomes in inoculum and in the serum at 45 min post-inoculation were quantified by RT-qPCR. Mean ± SD. N = 8, two experiments. Mann-Whitney test; p>0.05.

DOI: https://doi.org/10.7554/eLife.49163.024

The following source data is available for figure 8:

**Source data 1.** Raw data for *Figure 8B-D*.

DOI: https://doi.org/10.7554/eLife.49163.025

clearance of circulating CHIKV particles, it remains possible that MZM contribute to the clearance of this group of alphaviruses in intact mice. Further work is needed to evaluate whether KCs are required for the clearance of circulating CHIKV, ONNV, and RRV particles, or whether other macrophage populations can compensate in their absence.

Beyond the antiviral antibody response, the virus-host interactions that determine the magnitude and duration of vertebrate viremia following an arbovirus infection are poorly understood. In some cases, alphavirus particles interact directly with glycosaminoglycans (GAGs), such as heparan sulfate (*Bernard et al., 2000*; *Byrnes and Griffin, 1998*; *Gardner et al., 2011*; *Heil et al., 2001*; *Klimstra et al., 1998*; *Smit et al., 2002*), that are ubiquitously found on the surface of mammalian cells. Following serial passage in cell culture, alphaviruses frequently acquire adaptive mutations that enhance interactions between viral particles and GAGs. (*Bernard et al., 2000*; *Byrnes and Griffin, 1998*; *Heil et al., 2001*; *Klimstra et al., 1998*; *Silva et al., 2014*). Alphaviruses that have acquired enhanced in vitro GAG binding are rapidly cleared from the circulation of mice (*Bernard et al., 2000*; *Byrnes and Griffin, 2000*), suggesting that virion-GAG interactions are one mechanism that can determine the magnitude and duration of arboviral viremia. However, the extent to which specific GAGs contribute to clearance of circulating alphavirus particles has not been defined in vivo. Beyond GAGs, previous work demonstrated that opsonization of DENV particles by mannose binding lectin (MBL), which can neutralize particles via a C3-dependent mechanism, accelerated their clearance from the circulation of mice (*Fuchs et al., 2010*). However, we found that the innate clearance of circulating CHIKV and RRV was maintained in mice lacking C3 and in mice lacking antibodies, suggesting that the clearance of these circulating alphavirus particles occurs through a non-opsonic mechanism.

SRs are expressed on a variety of tissue resident macrophage populations, including those in the liver and spleen that surveil the blood (*Kubes and Jenne, 2018*; *Lewis et al., 2019*). SRs mediate non-opsonic clearance of a variety of circulating self and non-self molecules due to their capacity to recognize a diverse range of ligands, including modified endogenous proteins or lipoproteins, as well as bacterial surface proteins, lipopolysaccharide (LPS), and lipotechoic acid (LTA) (*Areschoug and Gordon, 2009*; *Canton et al., 2013*). We found that the clearance of circulating CHIKV and RRV particles was inhibited by pre-treatment of mice with the SR competitive inhibitors poly(I) and dextran sulfate (*Platt and Gordon, 1998*), supporting a role for SRs in clearance of these alphaviruses from the circulation. Poly(I) and dextran sulfate are known ligands for the class A scavenger receptors SR-A1 and MARCO (*Chen et al., 2006*; *Platt and Gordon, 1998*), both of which are transcriptionally expressed in liver KCs (*Tabula Muris Consortium et al., 2018*). SR-A1 and MARCO play a prominent role in host defense by functioning as phagocytic receptors for multiple bacterial pathogens. For example, mice deficient in SR-A1 are more susceptible to infection with *L. monocytogenes, S. aureus, S. pneumonia* and *N. meningitides,* and MARCO$^{-/-}$ mice have enhanced sensitivity to infection with *S. pneumonia* (*Areschoug and Gordon, 2009*). However, less is known about the role of class A scavenger receptors during viral infections. SR-A1 has a protective role during systemic herpes simplex virus type 1 (HSV-1) infection of mice (*Suzuki et al., 1997*), and HSV-1 can utilize MARCO for attachment and entry into epithelial cells (*MacLeod et al., 2013*). Similarly, vaccinia virus (VacV) binding to MARCO on keratinocytes enhances infection of these cells in vitro (*MacLeod et al., 2015*). MARCO also mediates transduction of macrophages by adenovirus in vitro, and appears to promote innate immune responses to the infection (*Maler et al., 2017*), suggesting that MARCO-virus interactions can regulate host cell responses to infection. Our studies in SR-A1$^{-/-}$ and MARCO$^{-/-}$ mice identified critical roles for MARCO in the efficient clearance of circulating CHIKV particles and for limiting viral dissemination and the magnitude and duration of viremia, suggesting that MARCO may limit disease severity following CHIKV infection.

Beyond identification of the cell type and specific receptor involved in the clearance of alphavirus particles from the circulation, we also defined viral features that facilitate host recognition and clearance of viral particles by phagocytic cells. N-linked glycosylation of DENV and WNV glycoproteins was previously found to influence clearance of these viral particles from the circulation of mice (*Fuchs et al., 2010*). However, we found that mutation of the predicted surface-exposed CHIKV E2 glycosylation sites had no impact on the clearance of CHIKV particles from the circulation, suggesting that clearance occurs independent of E2 glycosylation. To define the viral features that mediated clearance of circulating alphavirus particles, we identified CHIKV and RRV strains that evade phagocytic cell-mediated clearance. CHIKV E2 K200R was isolated from the serum of a *Rag1*$^{-/-}$ mouse chronically infected with an Asian genotype CHIKV strain (*Hawman et al., 2017*), suggesting that this mutation arose as a mechanism to evade phagocytic cell-mediated clearance from the circulation. Introduction of an E2 K200R mutation into CHIKV strains representative of the Asian-American, ECSA, and West African CHIKV clades also facilitated escape of viral particles from phagocytic cell-

mediated clearance and enhanced viral dissemination following subcutaneous inoculation. Moreover, E2 K200 is conserved in the closely related alphavirus ONNV, and introduction of an E2 K200R mutation in this independent alphavirus species also allowed for escape from clearance from the circulation. Collectively, these data demonstrate that an E2 K200R mutation can function in a variety of distinct genetic backgrounds to facilitate escape of circulating alphavirus particles from MARCO-dependent clearance. While we found that circulating RRV particles also are susceptible to clearance by phagocytic cells, RRV has an asparagine at E2 200, suggesting that a viral feature other than E2 K200 promotes the clearance of circulating RRV particles. Analysis of a panel of RRV strains revealed that the RRV T48 strain, in contrast to strains SN11 and DC5692, was relatively resistant to clearance by phagocytic cells. Through chimeric and mutational analysis of susceptible (DC5692) and resistant (T48) RRV strains, we discovered that a lysine residue at E2 251 is essential for efficient clearance of RRV particles from the circulation. Thus, discrete lysine residues in the E2 glycoprotein of three independent alphaviruses promote their efficient clearance from the circulation.

We were surprised that single lysine to arginine mutations at discrete sites in E2 allowed circulating alphavirus particles to escape from clearance by phagocytic cells, as the side chains of lysine and arginine are both positively charged and similar in length. To more thoroughly evaluate biochemical features of the residue at CHIKV E2 200 or RRV E2 251 that promote particle clearance or particle escape from clearance, we generated panels of CHIKV and RRV strains encoding amino acids at positions E2 200 or 251 with positively-charged, negatively-charged, hydrophobic, or polar uncharged side chains. Moreover, this panel of amino acids also had a wide variety of side chain lengths and structures. Remarkably, with the exception of lysine, all amino acid substitutions at CHIKV E2 200 or RRV E2 251 allowed for evasion of clearance from the circulation. These findings suggest that a lysine residue at CHIKV E2 200 or RRV E2 251 is essential for MARCO-dependent clearance of circulating viral particles. Taken together, these observations suggest that these lysine residues may be post-translationally modified to facilitate capture and clearance of circulating alphavirus particles. A number of observations from this study and previously published studies support this idea, including (1) an E2 K200R mutation functions in a variety of distinct genetic backgrounds to facilitate escape of circulating CHIKV and ONNV particles from MARCO-dependent clearance; (2) a lysine residue at a distinct site in RRV E2 is essential for phagocytic-cell mediated clearance of circulating RRV particles; (3) while the lysine that mediates RRV clearance is conserved in CHIKV, mutating this lysine has no impact on clearance of circulating CHIKV particles; (4) lysine is frequently post-translationally modified, and can be modified with a diverse range of post-translational modifications (PTMs) (*Azevedo and Saiardi, 2016*); and (5) SRs were initially identified based on their capacity to capture lysine acetylated low density lipoprotein (*Platt and Gordon, 1998*). However, whether specific features of unmodified lysine residues within particular structural contexts of E2, or post-translational modifications of these lysine residues, allow for alphavirus recognition by phagocytic cells requires additional biochemical and structural analyses, and thus remains to be determined.

Despite the apparent benefit for the virus to encode an arginine or other amino acid at E2 200 or 251, lysine residues at CHIKV E2 200, ONNV E2 200, and RRV E2 251 are commonly found in virus isolates. Analysis of E2 protein sequences in the ViPR database (*Pickett et al., 2012*) revealed that 1,333/1,335 CHIKV sequences encode a lysine at position E2 200 (2/1,335 encode an arginine), 8/8 ONNV sequences encode a lysine, and 187/204 available RRV sequences encode a lysine at position E2 251, while the remaining 17 encode an arginine. One possible explanation for this conservation could be that a lysine residue at this position provides a fitness advantage in the mosquito vector. However, we found that CHIKV E2 K200R replication and dissemination was indistinguishable from that of WT CHIKV in *Ae. aegypti* mosquitoes, and no selective disadvantage of an arginine residue at this position was observed in direct competition experiments, suggesting that the conservation of lysine at CHIKV E2 200 is not due to constraints in the mosquito vector. Alternatively, the susceptibility or resistance of circulating alphavirus particles to clearance may be influenced by polymorphisms in SRs. Genes encoding SRs, including MARCO, exhibit substantial genetic variation in both humans and mice (*Bowdish and Gordon, 2009*) and this variation can have functional consequences. For example, position 252 of MARCO is known to be polymorphic in humans, and the less common variant has a decreased capacity to phagocytize bacteria in vitro (*Novakowski et al., 2018*). Consistent with this, SNPs within human SRs associate with differential susceptibility to various pathogens (*Bowdish et al., 2013*; *High et al., 2016*; *Lao et al., 2017*; *Ma et al., 2011*; *Westhaus et al., 2017*). Following alphavirus infections, human patients exhibit a high degree of variability in peak viremia

and disease severity (*Chow et al., 2011*; *de St Maurice et al., 2018*; *Musso et al., 2017*; *Thiberville et al., 2013*; *Vaughn et al., 2000*; *Waggoner et al., 2016*). Given this, inter- or intra-host variability within SRs may influence the susceptibility of alphaviruses to SR-dependent clearance from the circulation, contributing to variability in viremia and disease outcomes. Similar species-specific effects have been observed with multiple pathogen recognition receptors (*Daugherty and Malik, 2012*). Furthermore, MARCO is evolving under positive selection (*Yap et al., 2015*), a signature often associated with host-viral conflicts. Thus, circulating alphaviruses may have adapted to their cognate hosts and can readily evade this defense mechanism in certain vertebrates, but the MARCO allele in the mouse strains evaluated in this study allows these alphaviruses to be efficiently cleared from the circulation. More work is needed to determine whether polymorphisms within or between vertebrate species influence SR-dependent clearance of circulating alphaviruses.

The efficient clearance of alphaviruses from the circulation by phagocytic cells has important implications for the course of infection. The CHIKV E2 K200R strain that escapes MARCO-dependent clearance generates a higher viremia, disseminates to distal sites more rapidly, and causes more severe disease outcomes in mice (*Hawman et al., 2017*). Analogously, infection of MARCO$^{-/-}$ mice with WT CHIKV also resulted in enhanced viremia and more rapid viral dissemination to distal sites, suggesting that the E2 K200R mutation allows the virus to escape from MARCO-dependent clearance. Consistent with our data, prior studies found that some strains of Venezuelan equine encephalitis virus accumulated in the liver following i.v. inoculation of hamsters (*Jahrling and Gorelkin, 1975*) and that depletion of phagocytic cells with clodronate liposomes resulted in a higher magnitude and duration of viremia following subcutaneous CHIKV infection of mice (*Gardner et al., 2010*). Similarly, clodronate liposome depletion of phagocytic cells enhanced WNV viremia and disease severity in mice (*Bryan et al., 2018*), suggesting that phagocytic cell-mediated clearance of circulating arboviruses has implications that span virus genera. Beyond disease outcomes, the magnitude and duration of viremia also greatly influences transmission cycles. While some arboviruses, such as WNV and Japanese Encephalitis virus, are unable to generate a viremia of sufficient magnitude in humans to infect a mosquito vector and propagate the transmission cycle (*Weaver, 2018*), others like CHIKV, DENV, and Zika virus are readily amplified through human-mosquito-human cycles (*Weaver, 2018*). Susceptibility or resistance to clearance by liver macrophages, other scavenging cell types such as liver sinusoidal endothelial cells, and SRs may contribute to whether humans are dead-end hosts for an arbovirus or can fuel widespread epidemics.

## Materials and methods

See *Supplementary file 2* for Key Resources Table.

### Ethics statement

This study was performed in strict accordance with the recommendations in the Guide for the Care and Use of Laboratory Animals of the National Institutes of Health. All of the animals were handled according to approved institutional animal care and use committee (IACUC) protocols (#00026) of the University of Colorado School of Medicine (Assurance Number A3269-01). Experimental animals were humanely euthanized at defined endpoints by exposure to isoflurane vapors followed by thoracotomy.

### Cells

Cell lines were obtained directly from ATCC. Vero cells (ATCC CCL81) were cultured in Dulbecco's Modified Eagle medium (DMEM)-F-12 (Life Technologies) supplemented with 10% fetal bovine serum (FBS, Lonza), 1x nonessential amino acids (Life Technologies), sodium bicarbonate, 2 mM L-glutamine, and penicillin-streptomycin. BHK-21 cells (ATCC CCL10) were cultured in α-minimum essential medium (Life Technologies) supplemented with 10% FBS (Lonza), 10% tryptone phosphate broth, and penicillin-streptomycin. Human dermal fibroblasts (ATCC PCS-201–012) were cultured in high-glucose DMEM supplemented with 10% FBS and penicillin-streptomycin. C6/36 cells (ATCC CRL-1660) were cultured in minimum essential medium with Earle's salts (Life Technologies) supplemented with 5% FBS (Lonza) and penicillin-streptomycin. All mammalian cells were cultured at 37°C, and C6/36 mosquito cells were incubated at 30°C. Cells tested negative for mycoplasma.

## Viruses

CHIKV strains used include Asian strain AF15561 (*Ashbrook et al., 2014*), Asian-American strain 99659 (*Jones et al., 2017*), ECSA stain SL15649 (*Morrison et al., 2011*), and WA strain 37997 (*Vanlandingham et al., 2005b*). All CHIKV stocks were derived from infectious cDNA clones by electroporating RNA into BHK-21 cells, as previously described (*Morrison et al., 2011*), with the exception of virus used in *Figure 4B*, which was a propagated P1 stock of AF15561 generated in human dermal fibroblasts. Unless otherwise noted, AF15561 was used as the representative CHIKV strain. RRV strains used include SN11 (*Liu et al., 2011*), T48 (*Doherty and Whitehead, 1963*; *Kuhn et al., 1991*), and DC5692 (*Jupille et al., 2011*; *Lindsay, 1996*). SN11 is a clinical isolate that was passed 1x on C6/36 cells before we propagated a stock in BHK-21 cells (*Liu et al., 2011*), while T48 and DC5692 were derived from cDNA clones by electroporation of viral RNA into BHK-21 cells as previously described (*Morrison et al., 2006*). For some experiments (*Figure 7—figure supplement 1C*), viral particles were propagated in serum free media and purified by centrifugation through a 20% sucrose cushion at 24,000 x g for 6 hr. Unless otherwise noted, SN11 was used as the representative RRV strain. ONNV SG650 was derived from a cDNA clone through electroporation into BHK-21 cells, and propagated on BHK-21 cells for one passage to increase titer. Mutant variants used were described previously, (AF15561 E2 K200R [*Hawman et al., 2017*]; T48-DC5692 E1/6K [RR73] [*Jupille et al., 2011*], T48-DC5692 E2/E3 [RR100] [*Jupille et al., 2011*], and T48 E2 R251K [*Jupille et al., 2013*]) or were generated through site-directed mutagenesis of cDNA clones. Mutant virus stocks were sequenced to confirm the presence of introduced mutations. All viruses were titered by plaque assay on BHK-21 cells or by RT-qPCR. The cDNA clone of CHIKV 99659 was a gift from Mark Heise (University of North Carolina), the cDNA clones of CHIKV 37997 and ONNV SG650 were gifts from Stephen Higgs (Kansas State University), and RRV SN11 was a gift from John Aaskov (Queensland University of Technology).

## Site directed mutagenesis

Viral point mutants were constructed through site-directed mutagenesis (Agilent #200517) of cDNA clones, using the primers provided in *Supplementary file 1*. A sequence verified fragment containing the desired mutation was then subcloned into the unmutagenized parental plasmid. For AF15561 E2 K200 mutants, SL15649 E2 K200R, 99659 E2 K200R, and AF15561 glycosylation mutants, subcloning was performed using restriction sites XhoI and XmaI. 37997 E2 K200R was not subcloned, but the E2 gene was sequence verified to only contain the K200R mutation. For ONNV E2 K200R, BstBI and BamHI were used for subcloning. RRV DC6592 E2 K251R was subcloned using site PspOMI and XmaI. RRV T48 mutants E2 R251A, E2 R251D, E2 R251Q, and E2 R251S were not subcloned, but the E2 gene was sequence verified to only contain the specified E2 251 mutation.

## Mouse experiments

BALB/cJ, 129S1/SvImJ, C57BL/6J, sham or splenectomized C57BL/6J, and congenic µMT (*Kitamura et al., 1991*), $C3^{-/-}$ (*Wessels et al., 1995*), and $SR-A1^{-/-}$ (*Suzuki et al., 1997*) mice were obtained from The Jackson Laboratory. $MARCO^{-/-}$ mice (*Arredouani et al., 2004*) were provided by Dawn Bowdish (McMaster University). All experiments were performed in 3–4 week-old mice, except those involving sham or splenectomized mice, which were 6 weeks-old at the time of virus inoculation. All WT mice used were male, with the exception of *Figure 2C*, where five mice were female. WT mice were purchased commercially and were age-matched, and distributed randomly across groups. For mouse strains bred in house (µMT, $C3^{-/-}$, $SR-A1^{-/-}$, $MARCO^{-/-}$), mice were distributed randomly into groups containing approximately equal division of the sexes. All experiments were performed under Biosafety level 2 or 3 conditions, as appropriate. Masking was not used. Phagocytic cell depletions were achieved by inoculating mice i.v. with 100 µl/10 g body weight of PBS- or clodronate-loaded liposomes (Clodronateliposomes.org) 42 hr prior to inoculation with virus. For clearance blockade experiments, mice were treated i.v. with 200 µg of poly(I) or poly(C) (Sigma 26936-41-4 and 26936-40-3) or 200 µg of dextran or dextran sulfate (Sigma 31392 and D6001) 5 min prior to inoculation with virus. For serum clearance experiments, mice were inoculated i.v. with a defined number of particles (CHIKV and RRV: $10^7$ or $10^8$ viral genomes depending on experiment; ONNV: $10^5$ PFU) diluted in a total volume of 100–200 µl of diluent (PBS with 1% FBS). Viral inoculum was reserved to quantify input genomes. At the time of termination, mice were euthanized through

inhalation of isoflurane vapors, followed by thoracotomy, and serum was collected. In some experiments, mice were then perfused by intracardiac injection of PBS prior to collecting the indicated tissues in RPMI for flow cytometry or in TRIzol Reagent (Life Technologies) for RNA analysis. Tissues for RNA analysis were first homogenized with a MagNA Lyser (Roche). For viral dissemination experiments, mice were inoculated in the left rear footpad with 10 µl containing $10^3$ PFU of virus diluted in PBS with 1% FBS. At 24 hpi, mice were euthanized and blood was collected. Mice were then perfused by intracardiac injection of PBS, and the indicated tissues were collected in in vitro diluent (PBS with 1% FBS, $1 \times Ca^{2+}$, and $1 \times Mg^{2+}$) and homogenized with a MagNA Lyser (Roche) for analysis by focus formation assay or plaque assay.

## RT-qPCR genome quantification

To quantify viral genomes, RNA was isolated from 20 µl of serum or homogenized tissues in TRIzol using the PureLink RNA mini kit (Life Technologies). CHIKV cDNA was generated using random primers (Invitrogen) with SuperScript IV reverse transcriptase (Life Technologies). CHIKV genome copies were quantified by RT-qPCR using a CHIKV specific forward primer (5′-TTTGCGTGCCACTCTGG-3′) and reverse primer (5′-CGGGTCACCACAAAGTACAA-3′) with an internal TaqMan probe (5′-ACTTGCTTTGATCGCCTTGGTGAGA-3′), as previously described (*Hawman et al., 2013*). To generate RRV cDNA, a sequence-tagged (indicated with lower case letters) RRV-specific RT primer (5′-ggcagtatcgtgaattcgatgcAACACTCCCGTCGACAACAGA-3′) was used with SuperScript IV reverse transcriptase (Life Technologies). To quantify RRV genomes by RT-qPCR, a tag sequence-specific reverse primer (5′-GGCAGTATCGTGAATTCGATGC-3′) was used with a RRV sequence-specific forward primer (5′-CCGTGGCGGGTATTATCAAT-3′) and an internal TaqMan probe (5′-ATTAAGAGTGTAGCCATCC-3′), as previously described (*Stoermer et al., 2012*).

## Plaque assay and focus formation assay

In some experiments, infectious virus was quantified by plaque assay or focus formation assay, as described previously (*Hawman et al., 2017*). In brief, for plaque assays, BHK-21 cells were seeded in a 6-well plate and inoculated with 10-fold serial dilutions of serum or tissue homogenate in in vitro diluent. Following a 1 hr adsorption, cells were overlaid with 1% immunodiffusion agarose (MP Biomedical) and incubated at 37°C for 40–44 hr before staining with neutral red stain to enumerate plaques. For focus formation assays, Vero cells were seeded in a 96-well plate and serial dilutions of serum or tissue homogenate were adsorbed to the cells for 2 hr, after which the inoculum was removed and an overlay of 0.5% methylcellulose in MEM-alpha with 10% FBS was added to the cells. Following an 18 hr incubation at 37°C, the cells were fixed with 1% paraformaldehyde and probed with CHK-11 monoclonal antibody at 500 ng/ml in Perm Wash (1x PBS, 0.1% saponin, 0.1% BSA), followed by a secondary goat anti-mouse IgG conjugated to horseradish peroxidase at 1:2000 in Perm Wash. Foci were visualized with TrueBlue substrate (Fisher) and counted with a CTL Biospot analyzer using Biospot software (Cellular Technology).

## In situ hybridization and immunohistochemistry

WT or MARCO$^{-/-}$ C57BL/6 mice were treated i.v. with 100 µl/10 g body weight of PLL or CLL 42 hr prior to i.v. inoculation with $2.5 \times 10^9$ genomes of CHIKV AF15561 diluted in 100 µl of diluent. At 45 min post-inoculation, mice were sacrificed, perfused with 4% paraformaldehyde (PFA), and liver lobes were collected and fixed in 4% PFA for 24 hr. Tissues were paraffin-embedded and sectioned. RNA in situ hybridization was performed using the RNAscope 2.5 HD Assay-Brown (Advanced Cell Diagnostics (ACD) Cat. No. 322300) according to the manufacturer's protocol. In brief, paraffin-embedded tissue sections were incubated at 60°C for 1 hr followed by deparaffinization. Endogenous peroxidase activity was quenched by pretreatment with hydrogen peroxide for 10 min at room temperature (RT), and slides were boiled in RNAscope Target Retrieval Solution for 15 min. Slides were then treated with Protease III at 40°C for 30 min in the ACD Ez II Hybridization Oven (ACD Cat. No. 321710), followed by hybridization of V-CHIKV-sp-01 probe (ACD Cat. No. 481891) for 2 hr at 40°C in the EZ II Oven. Following hybridization of AMP1-6, signal was detected using DAB substrate and tissues were counterstained with Gill's hematoxylin and visualized using standard bright-field microscopy. On sections from the same liver lobes, immunohistochemistry was used to visualize F4/80 positive cells. Tissues were deparaffinized and target retrieval was performed by incubating slides

in Citrate pH 6.1 Target Retrieval Solution (Dako Cat. No. S1699) in a Biocare Medical decloaking chamber at 110°C for 20 min. Tissues were then incubated with Dual Endogenous Enzyme Block (Dako Cat. No. S2000389) for 10 min at RT, followed by incubation with Protein Block (Dako Cat. No. X090930) for 10 min at RT. F4/80 antibody clone Cl:A3-1 (BioRad Cat. No. MCA497) was diluted 1:100 and added to tissues and incubated for 1 hr at RT, followed by biotinylated Anti-Rat IgG (Vector Laboratories BA-9401) for 30 min at RT. Tissues were then incubated with VECTASTAIN Elite ABS reagent (Vector Laboratories, PK-6100) at RT for 30 min, signal was detected using DAB substrate, and tissues were counterstained with Gill's hematoxylin and imaged. To quantify F4/80 positive cells in WT and MARCO$^{-/-}$ livers, positively-stained cells in 10 randomly selected high power fields per section were counted in a blinded manner and used to calculate the average number of F4/80 positive cells per field.

## Thermostability assays

$10^6$ PFU of virus (CHIKV WT or E2 K200R, AF15561 strain; RRV T48 or T48 E2 R251K) were incubated in triplicate in normal mouse serum (Thermo Fisher Scientific #10410) at either 4°C or 37°C. Aliquots from 0, 1, 8, 24 and 48 hr time points were titered by plaque assay (RRV) or Focus Formation Assay (CHIKV), as described above.

## Flow cytometry

Livers were minced and incubated in digestion buffer (RPMI 1640 with 10% FBS, 15 mM HEPES, 2.5 mg/ml Collagenase Type 1 (Worthington Biochemical), 1.7 mg/ml DNase I (Roche), 1% penicillin/ streptomycin) for 30 min at 37°C with shaking (130 rpm). Digested livers were passed through a 100 μm cell strainer (BD Falcon). Liver cells were pelleted and re-suspended in a 35% Percoll solution and centrifuged for 15 min at 450 x g at 4°C with medium acceleration and no break to isolate lymphocytes, and washed 1X with RPMI. Isolated cells from livers were blocked with anti-mouse FcγRII/ III (2.4G2; BD Pharmingen) for 20 min on ice and then stained in FACS buffer (1x PBS, 2% FBS) for 1 hr on ice with the following antibodies from Biolegend (most) or eBioscience (NK1.1): anti-MHC-II (M5/114.14.2), anti-Ly6C (HK1.4), anti-F4/80 (BM8), anti-CD11b (M1/70), anti-NK1.1 (PK136), anti-TCRB (H57-597), anti-CD19 (6D5), anti-CD11c (N418), anti-CD45 (30-F11), anti-Ly6G (1A8). Liver cells were fixed in 1% paraformaldehyde (PFA) for 15 min. Samples were analyzed on an LSRFortessa using FACSDiva software (Becton Dickinson). Further analysis was done using FlowJo Software (Tree Star).

## Mosquito vector competence studies

 *Aedes aegypti* mosquitoes from Mexico (Poza Rica [*Vera-Maloof et al., 2015*], F18-20) were used for vector competence studies. Mosquitoes were fed citrated sheep blood for colony maintenance and provided with sugar and water ad libitum. Larvae and adults were reared and maintained under controlled conditions of 28°C temperature, 70% relative humidity and a 12:12 (L:D) diurnal light cycle. Mosquitoes were provided an infectious blood meal containing either CHIKV AF15561 WT or CHIKV AF15561 E2 K200R. Defibrinated calf blood was mixed 1:1 with DMEM containing $2.2 \times 10^6$ PFU/mL virus, resulting in a final blood meal titer of $1.1 \times 10^6$ PFU/mL. On day 3 post-infection, mosquitoes were cold-anesthetized and legs and wings were removed and placed into a 2 mL tube containing 250 μL mosquito diluent (1x PBS containing 20% FBS, antibiotics and antifungals) and a sterile stainless-steel bead. Mosquito saliva was then collected by placing the mosquito proboscis into a capillary tube filled with immersion oil to salivate for 30 min. Each capillary tube end containing saliva was broken off into a microcentrifuge tube containing 100 μL mosquito diluent, centrifuged for 3 min at 15,000 x g and stored at −80°C. After salivation, mosquito bodies were also placed in a 2 mL tube containing 250 μL mosquito diluent and a sterile stainless-steel bead. Tissues (legs/wings and bodies) were homogenized using a Retsch Mixer Mill MM400 (Germany) at 24 cycles/second for 1 min, centrifuged at 15,000 x g for 3 min and stored at −80°C. To determine whether samples contained virus or not, Vero cells grown in 24-well plates were inoculated with 50 μL mosquito sample for 1 hr before a semi-solid overlay (0.6% Tragacanth gum in EMEM) was added. Cells were fixed and stained with crystal violet three dpi to visualize plaques. Samples found to be positive for virus were further evaluated by focus formation assay, as described above, to quantify the amount of infectious virus present in the sample.

## Viral competition assays

To perform viral competition assays, we generated a genetically marked clone of CHIKV AF15561 using sight- directed mutagenesis (Forward Primer: 5′-ctaaactaaaggggcccaaagcagcagcgctgt-3′; Reverse Primer: 5′-acagcgctgctgctttgggcccctttagtttag-3′) to introduce a silent mutation that generates an ApaI restriction site in nsP4, as described previously (*Chen et al., 2013*). For in vitro competition experiments, C6/36 cells were seeded in 24-well dishes and inoculated in triplicate at an MOI of 1 PFU/cell with a 1:1 ratio of CHIKV AF15561-ApaI and CHIKV AF15561 or CHIKV AF15561-ApaI and CHIKV AF15561 E2 K200R. Following a 1 hr adsorption, virus inoculum was removed, cells were washed, and fresh media was added. At 24 hpi, 1/10th of the supernatant was passaged onto a new well of cells, and this was repeated for a total of five passages. RNA was extracted from the supernatant, and cDNA was generated as described above for CHIKV quantification. For in vivo competition experiments, *Aedes aegypti* mosquitoes were microinjected intrathoracically with 138 PFU of a 1:1 mixture of CHIKV AF15561-ApaI and CHIKV AF15561 or CHIKV AF15561-ApaI and CHIKV AF15561 E2 K200R. At seven dpi, mosquito bodies and saliva were collected and processed as described above and frozen at −80°C. RNA was extracted from 50 µL of each sample using the Mag-Bind Viral DNA/RNA 96 kit (Omega Bio-Tek) on the KingFisher Flex Magnetic Particle Processor (Thermo Fisher Scientific) following the manufacturer's instructions. cDNA was generated as described above for CHIKV quantification. Relative amounts of each virus in samples were evaluated as described previously (*Chen et al., 2013*). In brief, cDNA was subjected to PCR using forward primer (5′-atatctaga-catggtgga-3′) and reverse primer (5′- tatcaaaggaggctatgtc-3′) to amplify the region of nsP4 (6106–6794) containing the ApaI site. PCR products were digested with equal amounts of ApaI and PspOMI (isoschizomers) at room temperature for 30 min, and at 37°C for 4 hr. Complete digestion was confirmed using control PCR products amplified from cDNA plasmid clones of CHIKV AF15561 WT or CHIKV AF15561-ApaI. Digested products were run on a 1% TAE gel, stained with ethidium bromide (Sigma), and imaged using Syngene G:Box. The percent band intensity was quantified using Syngene GeneTools.

## Statistical analysis

To determine the sample size to be used, population variance from pre-existing sample sets were used in a power calculation (80% power, 0.05 type I error) to detect a 4–5-fold effect. The defined sample size (N) listed in each figure legend refers to biological replicates of individual mice, mosquitoes, or cell wells. Data are represented as mean ± SD. The statistical tests used for each data set are indicated in the figure legends. Statistical analysis was conducted using GraphPad Prism 7.0. Two-sided t-tests (parametric) or Mann Whitney tests (nonparametric) were used to compare two groups. One-way ANOVA with Bonferroni's multiple comparison test (parametric) or Kruskal-Wallis with Dunn's multiple comparisons test (nonparametric) were used to compare three or more groups, and two-way ANOVA with Bonferroni's multiple comparison test was used to compare two groups at multiple time points.

## Acknowledgements

This work was supported by Public Health Service grants R01 AI123348 to TEM, F32 AI140567 to KSC, and T32 AI007405 to KSC from the National Institute of Allergy and Infectious Diseases.

## Additional information

### Funding

| Funder | Grant reference number | Author |
| --- | --- | --- |
| National Institute of Allergy and Infectious Diseases | R01 AI123348 | Thomas E Morrison |
| National Institute of Allergy and Infectious Diseases | F32 AI140567 | Kathryn S Carpentier |
| National Institute of Allergy and Infectious Diseases | T32 AI007405 | Kathryn S Carpentier |

The funders had no role in study design, data collection and interpretation, or the decision to submit the work for publication.

### Author contributions
Kathryn S Carpentier, Conceptualization, Formal analysis, Funding acquisition, Investigation, Visualization, Methodology, Writing—original draft, Writing—review and editing; Bennett J Davenport, Kelsey C Haist, Nicholas A May, Alexis Robison, Investigation, Writing—review and editing; Mary K McCarthy, Formal analysis, Investigation, Methodology, Writing—review and editing; Claudia Ruckert, Investigation, Methodology, Writing—review and editing; Gregory D Ebel, Resources, Supervision, Methodology, Writing—review and editing; Thomas E Morrison, Conceptualization, Resources, Supervision, Funding acquisition, Visualization, Methodology, Writing—original draft, Project administration, Writing—review and editing

### Author ORCIDs
Kathryn S Carpentier (iD) https://orcid.org/0000-0003-0829-4544
Thomas E Morrison (iD) https://orcid.org/0000-0002-1811-2938

### Ethics
Animal experimentation: This study was performed in strict accordance with the recommendations in the Guide for the Care and Use of Laboratory Animals of the National Institutes of Health. All of the animals were handled according to approved institutional animal care and use committee (IACUC) protocols (#00026) of the University of Colorado School of Medicine (Assurance Number A3269-01). Experimental animals were humanely euthanized at defined endpoints by exposure to isoflurane vapors followed by thoracotomy.

### Decision letter and Author response
Decision letter https://doi.org/10.7554/eLife.49163.032
Author response https://doi.org/10.7554/eLife.49163.033

## Additional files
### Supplementary files
• Supplementary file 1. Primers used to generate mutant viruses through site-directed mutagenesis.
DOI: https://doi.org/10.7554/eLife.49163.026
• Supplementary file 2. Key Resources Table.
DOI: https://doi.org/10.7554/eLife.49163.027
• Transparent reporting form DOI: https://doi.org/10.7554/eLife.49163.028

### Data availability
All data generated or analysed during this study are included in the manuscript and supporting files.

The following previously published dataset was used:

| Author(s) | Year | Dataset title | Dataset URL | Database and Identifier |
|---|---|---|---|---|
| The Tabla Muris Consortium | 2018 | Tabula Muris: Transcriptomic characterization of 20 organs and tissues from Mus musculus at single cell resolution | https://www.ncbi.nlm.nih.gov/geo/query/acc.cgi?acc=GSE109774 | Gene Expression Omnibus, GSE109774 |

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
