## [Decision Letter]

Thank you for submitting your article "Discrete lysines in the E2 glycoprotein mediate capture of circulating Alphavirus particles via MARCO" for consideration by *eLife*. Your article has been reviewed by three peer reviewers, one of whom is a member of our Board of Reviewing Editors, and the evaluation has been overseen by Karla Kirkegaard as the Senior Editor. The reviewers have opted to remain anonymous.

The reviewers have discussed the reviews with one another, and the Reviewing Editor has drafted this decision to help you prepare a revised submission.

Summary:

In this study, the authors demonstrate an important role for liver Kupffer cells in the clearance of alphaviruses from the circulation of infected mice. Clearance requires scavenger receptors and studies in knockout mice indicate that MARCO might be the KC receptor responsible for alphavirus clearance from the circulation. Further, the authors characterize the specific lysine residues in the viral E2 glycoproteins that are critical for viral clearance.

Essential revisions:

Several concerns raised by the reviewers should be addressed before the paper can be accepted.

1) There is no evidence of direct interaction of these viruses with MARCO, nor of the role of lysine in such an interaction. The most important evidence would be in vitro experiments showing differential viral binding to matched MARCO+ and MARCO- cells. Both WT and lysine-mutant viruses should be tested for their interaction with these cells. It's also possible that similar studies could be performed in vivo by showing co-localization of viral RNA with KC in WT livers compared to KO. However, the reviewers acknowledge that such in vivo experiments may take much longer than the requested 2 month time frame and are therefore lower in priority.

2) To consider alternative hypotheses, the authors should evaluate whether KC in MARCO KO mice are present in normal numbers, have normal function (uptake of latex beads, for example), and express other scavenger receptors at normal levels, particularly since SR knockout mice sometimes show compensatory effects on expression of other proteins (Mol Ther 2013, 21:767, PMID: 23358188).

3) Title: In its current form, the title suggests MARCO directly interacts with lysines in E2, which is premature until the Essential Revision #1 is addressed. Similarly, conclusions elsewhere in the manuscript are also overly strong in the absence of this data.

---

## [Author Response]

Essential revisions:Several concerns raised by the reviewers should be addressed before the paper can be accepted.1) There is no evidence of direct interaction of these viruses with MARCO, nor of the role of lysine in such an interaction. The most important evidence would be in vitro experiments showing differential viral binding to matched MARCO+ and MARCO- cells. Both WT and lysine-mutant viruses should be tested for their interaction with these cells. It's also possible that similar studies could be performed in vivo by showing co-localization of viral RNA with KC in WT livers compared to KO. However, the reviewers acknowledge that such in vivo experiments may take much longer than the requested 2-month time frame and are therefore lower in priority.

To address these reviewer concerns, we i.v. pretreated WT and MARCO^-/-^ mice with PLL or CLL. At 42 hours post-treatment, mice were i.v. inoculated with CHIKV. At 45 min post virus inoculation the following analyses were performed: (1) viral RNA in the serum was quantified by RT-qPCR, (2) viral RNA accumulation in the liver was assessed by in situ hybridization, and (3) F4/80^+^ cells in the liver were visualized by immunohistochemistry and quantified by counting the number of F4/80^+^ cells per field. These new analyses demonstrated that pretreatment with CLL blocked clearance of circulating CHIKV particles in WT mice, but had no effect on the level of CHIKV in the circulation of MARCO^-/-^ mice (new Figure 3G). Furthermore, CHIKV RNA was detectable by in situ hybridization in the liver of PLL-treated WT mice, but not in the liver of PLL-treated MARCO^-/-^ mice or CLL-treated WT and MARCO^-/-^ mice (new Figure 3H). In addition, despite the virological differences in the circulation and liver of PLL-treated WT and MARCO^-/-^ mice, these mice had a similar number of KCs in the liver (new Figure 3I). Finally, both WT and MARCO^-/-^ KCs were efficiently depleted by CLL (new Figure 3H and 3I), indicating that MARCO^-/-^ KCs retain the capacity to phagocytose liposomes. Thus, these new experiments support the conclusions that MARCO is required for clearance of circulating CHIKV particles and that MARCO is required for accumulation of CHIKV in the liver following i.v. inoculation of viral particles. In addition, these experiments address concerns raised in point #2 below. That is, the new data show that WT and MARCO^-/-^ mice have a similar number of F4/80^+^ KCs in the liver and that in both strains of mice these cells are efficiently depleted by CLL, which requires functional phagocytic processes.

We appreciate the reviewers request for viral binding studies in matched MARCO+ and MARCO- cells. However, viral binding to cells in culture is influenced by a variety of factors, including cell type and the abundance and nature of a number of other attachment factors such as glycosaminoglycans and proteinaceous entry receptors (Mxra8 and one or more additional unknown receptors) (1, 2). Moreover, flow-induced shear forces that occur in liver sinusoids (the anatomical home of KCs) may promote the interaction between alphaviral particles and MARCO. This concept has been formally demonstrated for capture of leukocytes by selectins expressed on endothelial cells. That is, their ligand-binding is improved under flow conditions (3, 4). In addition, a similar process has recently been described for neutrophil arrest; flow conditions dramatically enhanced neutrophil arrest by the CD99 receptor PILR-β1, which displays low affinity for CD99 under static conditions (5). Thus, while we are currently developing cell-based assays to explore a role for MARCO in viral binding, including the use of primary Kupffer cells under flow conditions, we feel that these complex experiments are beyond the scope of the current study.

2) To consider alternative hypotheses, the authors should evaluate whether KC in MARCO KO mice are present in normal numbers, have normal function (uptake of latex beads, for example), and express other scavenger receptors at normal levels, particularly since SR knockout mice sometimes show compensatory effects on expression of other proteins (Mol Ther 2013, 21:767, PMID: 23358188).

As outlined above, we performed new experiments to address these reviewer concerns. We found that WT and MARCO^-/-^ mice have a similar number of F4/80^+^ KCs in the liver and that KCs are efficiently depleted by CLL, which requires functional phagocytic processes, in both strains of mice. These new data are displayed in new Figure 3H and 3I.

As the reviewers highlight, in some cases the lack of one SR is not sufficient to observe an effect due to functional compensation by other SRs. For example, in the study cited by the reviewers, in contrast to poly(I) treatment, genetic deletion of SR-A1 did not enhance transduction of hepatocytes with adenovirus. Instead, the authors found that adenovirus also interacted with SREC-I, and that SREC-I expression was increased in the liver of SR-A1^-/-^ mice. Thus, antibody-mediated blockade of both SR-A1 and SREC-I mimicked the effects of treatment with poly(I) on adenovirus transduction. However, in our studies, the level of CHIKV in the circulation (i.e., the inhibitory effect on CHIKV clearance) is similar in WT mice treated with poly(I) or CLL and MARCO^-/-^ mice (Figure 3). Thus, our data suggest that other SRs are unable to compensate for the genetic absence of MARCO.

3) Title: In its current form, the title suggests MARCO directly interacts with lysines in E2, which is premature until the Essential Revision #1 is addressed. Similarly, conclusions elsewhere in the manuscript are also overly strong in the absence of this data.

We appreciate these reviewer concerns. To address these concerns we have made the following modifications to the text of the manuscript:

1) The title of the manuscript has been changed to “Discrete Viral E2 Lysine Residues and Scavenger Receptor MARCO are Required for Clearance of Circulating Alphaviruses”.

2) We changed “These findings suggest that blood-exposed phagocytic cells, such as those in the spleen or liver, efficiently capture blood-borne alphaviruses” to read “These findings suggest that blood-exposed phagocytic cells, such as those in the spleen or liver, are required for clearance of blood-borne alphaviruses”.

3) We changed “To confirm that the enhanced viremia and dissemination associated with the CHIKV E2 K200R mutant viruses was due to an interaction with MARCO,…” to read “To investigate if expression of MARCO also influences viremia and viral dissemination following subcutaneous inoculation of virus,…”.

4) We changed “Collectively, these findings suggest that MARCO-mediated capture of circulating CHIKV particles controls viremia and impedes viral dissemination” to read “Collectively, these findings indicate that MARCO-dependent clearance of circulating CHIKV particles controls viremia and impedes viral dissemination”.

5) We changed “suggesting that the E2 K200R mutation disrupts viral interactions with MARCO” to “suggesting that the E2 K200R mutation allows the virus to escape from MARCO-dependent clearance”.

6) The title of Figure 3 was changed from “The scavenger receptor MARCO mediates clearance of CHIKV and RRV” to “The scavenger receptor MARCO is required for clearance of CHIKV and RRV”.

5) Throughout the text, we changed instances of “MARCO-mediated” or “SR-mediated” to “MARCO-dependent” or “SR-dependent”.

REFERENCES

1. Silva LA, Khomandiak S, Ashbrook AW, Weller R, Heise MT, Morrison TE, Dermody TS. 2014. A single-amino-acid polymorphism in Chikungunya virus E2 glycoprotein influences glycosaminoglycan utilization. J Virol 88:2385-2397.

2. Zhang R, Kim AS, Fox JM, Nair S, Basore K, Klimstra WB, Rimkunas R, Fong RH, Lin H, Poddar S, Crowe JE, Jr., Doranz BJ, Fremont DH, Diamond MS. 2018. Mxra8

is a receptor for multiple arthritogenic alphaviruses. Nature 557:570-574.

3. Marshall BT, Long M, Piper JW, Yago T, McEver RP, Zhu C. 2003. Direct observation of catch bonds involving cell-adhesion molecules. Nature 423:190-193.

4. Yago T, Wu J, Wey CD, Klopocki AG, Zhu C, McEver RP. 2004. Catch bonds govern adhesion through L-selectin at threshold shear. J Cell Biol 166:913-923.

5. Li YT, Goswami D, Follmer M, Artz A, Pacheco-Blanco M, Vestweber D. 2019. Blood flow guides sequential support of neutrophil arrest and diapedesis by PILR-beta1 and PILR-alpha. Elife 8.

6. Van Rooijen N. 1989. The liposome-mediated macrophage 'suicide' technique. J Immunol Methods 124:1-6.